
1       Aerosol-radiation feedback deteriorates the wintertime haze in North China Plain

Jiarui Wu[1,6], Naifang Bei[2], Bo Hu[3], Suixin Liu[1], Meng Zhou[4], Qiyuan Wang[1], Xia Li[1,6], Lang Liu[1], Tian Feng[1],
4       Zirui Liu[3], Yichen Wang[1], Junji Cao[1], Xuexi Tie[1], Jun Wang[4], Luisa T. Molina[5], and Guohui Li[1*]
[1]Key Lab of Aerosol Chemistry and Physics, SKLLQG, Institute of Earth Environment, Chinese Academy of
Sciences, Xi'an, Shaanxi, China
[2]School of Human Settlements and Civil Engineering, Xi'an Jiaotong University, Xi'an, Shaanxi, China
[3]State Key Laboratory of Atmospheric Boundary Layer Physics and Atmospheric Chemistry, Institute of
Atmospheric Physics, Chinese Academy of Sciences, Beijing, 100029, China
[4]Department of Chemical and Biochemical Engineering & Interdisciplinary Graduate Program in
Geo-Informatics, University of Iowa, Iowa City, Iowa, USA
[5]Molina Center for Energy and the Environment, La Jolla, California, USA
[6]University of Chinese Academy of Science, Beijing, China
[*]Correspondence to: Guohui Li (ligh@ieecas.cn)
**Abstract**. Atmospheric aerosols or fine particulate matters (PM$_{2.5}$) scatter or absorb a fraction
of the incoming solar radiation to cool or warm the atmosphere, decreasing surface
temperature and altering atmospheric stability to further affect the dispersion of air pollutants
in the planetary boundary layer (PBL). In the present study, simulations during a persistent
and heavy haze pollution episode from 05 December 2015 to 04 January 2016 in the North
China Plain (NCP) were performed using the WRF-CHEM model to comprehensively
quantify contributions of the aerosol shortwave radiative feedback (ARF) to near-surface
PM$_{2.5}$ mass concentrations. The WRF-CHEM model generally performs well in simulating
the temporal variations and spatial distributions of air pollutants concentrations compared to
observations at ambient monitoring sites in NCP, and the simulated diurnal variations of
aerosol species are also consistent with the measurements in Beijing. Additionally, the model
simulates well the aerosol radiative properties, the downward shortwave flux, and the PBL
height against observations in NCP during the episode. During the episode, the ARF
deteriorates the haze pollution, increasing the near-surface PM$_{2.5}$ concentration in NCP by
10.2 μg m$^{-3}$ or with a contribution of 7.8% on average. Sensitivity studies have revealed that
high loadings of PM$_{2.5}$ during the episode attenuate the incoming solar radiation down to the
surface, cooling the temperature of the low-level atmosphere to suppress development of
PBL and decrease the surface wind speed, further enhancing the relative humidity and
hindering the PM$_{2.5}$ dispersion and consequently exacerbating the haze pollution in NCP. The
ensemble analysis indicates that when the near-surface PM$_{2.5}$ mass concentration increases
from around 50 to several hundred μg m$^{-3}$, the ARF contributes to the near-surface PM$_{2.5}$ by
more than 20% during daytime in NCP, substantially aggravating the heavy haze formation.
However, when the near-surface PM$_{2.5}$ concentration is less than around 50 μg m$^{-3}$, the ARF



generally reduces the near-surface $PM_{2.5}$ concentration due to the consequent perturbation of
atmospheric dynamic fields.



## 1 Introduction

Atmospheric aerosols, produced both naturally and anthropogenically, influence the
radiative energy budget of the Earth's atmospheric system in many ways. They scatter or
absorb a fraction of the incoming solar radiation to cool or warm the atmosphere, decreasing
surface temperature and altering atmospheric stability (e.g., Ackerman, 1977; Jacobson, 1998,
2002). Also, they serve as cloud condensation nuclei (CCN) and ice nuclei (IN), thus
modifying cloud optical properties and lifetime (e.g., Zhang et al., 2007; Li et al., 2008;
2009). Among those impacts, the scattering and absorption of solar radiation by aerosols and
the associated feedbacks (hereafter referred to as aerosol-radiation feedback or ARF) not only
constitute one of the main uncertainties in climate prediction (IPCC, 2007), but also
substantially affect the atmospheric chemistry by perturbing the temperature profile and
moistures, winds, and planetary boundary layer (PBL) stability (Boucher et al., 2013).
Particularly, as a short-lived pollutant with uneven distribution and physical and chemical
heterogeneities in the atmosphere, the ARF varies by more than a factor of ten with location
or time of emissions (Penner et al., 2010).
During wildfire with high loading absorbing aerosols, the ARF has been reported to heat
the atmosphere and cool the surface, and thence enhance the PBL stability (e.g., Grell et al.,
2011; Fu et al., 2012; Wong et al., 2012). In addition, numerous studies have been performed
to evaluate impacts of the ARF of dust on the regional meteorology and climate (e.g., Perez et
al., 2006; D. Zhang et al., 2009; Santese et al., 2010). Anthropogenic aerosols, dominated by
scattering components, such as organics and sulfate, primarily attenuate the incoming solar
radiation down to the surface, cooling the temperature of the low-level atmosphere to
suppress the development of PBL and hinder the aerosol dispersion in the vertical direction
(e.g., Fast et al., 2006; Vogel et al., 2009; Zhang et al., 2010). In addition, the temperature
profile perturbation caused by the ARF also alters cloud formation and development, possibly



causing the precipitation delay or decrease (e.g., Zhao et al., 2005; Koch and Del Genio, 2010;
Ding et al., 2013).
Rapid industrialization and urbanization in China have significantly elevated the
concentrations of aerosols or fine particulate matters ($PM_{2.5}$), causing frequent occurrence of
haze pollution, particularly during wintertime in North China (e.g., Zhang et al., 2013; Pui et
al., 2014). Guo et al. (2014) have elucidated the haze formation mechanism in China,
highlighting the efficient aerosol nucleation and growth during haze episodes. Moreover, high
loading aerosols during heavy haze episodes induce efficient ARF, encumbering the PBL
development and further deteriorating the haze pollution. It is worth noting that the ARF
increases precursors for the aerosol nucleation and growth in the PBL, such as sulfuric and
organic gases, causing efficient aerosol nucleation and growth (Zhang et al., 2004; Guo et al.,
2014). Based on field measurements, recent studies have proposed that the high level of
$PM_{2.5}$ increases the stability of PBL due to the ARF and further decrease the PBL height
(PBLH), consequently enhancing $PM_{2.5}$ concentrations ($[PM_{2.5}]$) (Quan et al., 2013; Petaja et
al., 2016; Yang et al., 2016; Tie et al., 2017; Ding et al., 2017). Online-coupled meteorology
and chemistry models have also been used to verify the impact of the ARF on the PBLH and
near-surface $[PM_{2.5}]$ during heavy haze episodes in North China (Z. Wang et al., 2014; Wang
et al., 2015; Zhang et al., 2015; Gao et al., 2015). However, the ARF impact on near-surface
$[PM_{2.5}]$ varies, depending on the evaluation time and location (Table 1). For example, the
two-way coupled WRF-CMAQ system has been employed to evaluate the ARF contribution
to the haze formation in January 2013 over the North China Plain (NCP), showing that the
ARF reduces the PBLH by 100 m and enhances near-surface $[PM_{2.5}]$ by up to 140 $\mu g \ m^{-3}$ in
Beijing (J. Wang et al., 2014). Therefore, it is still imperative to comprehensively quantify the
ARF contribution to near-surface $[PM_{2.5}]$ under various pollution levels to provide the
underlying basis for supporting the design and implementation of emission control strategies.



In this study, simulations are performed using the Weather Research and Forecast model
with Chemistry (WRF-CHEM) to interpret the relationship between the near-surface $[PM_{2.5}]$
and the PBLH and further quantify the ARF contribution to near-surface $[PM_{2.5}]$ under
various pollution levels. The model and methodology are described in Section 2. Analysis
results and discussions are presented in Section 3, and summary and conclusions are given in
Section 4.
**2    Model and methodology**
**2.1   WRF-CHEM model and configurations**
The WRF-CHEM model (Grell et al., 2005) with modifications by Li et al. (2010, 2011a,
b, 2012) is applied to evaluate effects of the ARF on the wintertime haze formation in NCP.
The model includes a new flexible gas phase chemical module and the CMAQ aerosol
module developed by US EPA (Binkowski and Roselle, 2003). The wet deposition is based
on the method in the CMAQ module and the dry deposition of chemical species follows
Wesely (1989). The photolysis rates are calculated using the FTUV (fast radiation transfer
model) with the aerosol and cloud effects on photolysis (Li et al., 2005, 2011a). The
inorganic aerosols are predicted using ISORROPIA Version 1.7, calculating the composition
and phase state of an ammonium-sulfate-nitrate-water inorganic aerosol in thermodynamic
equilibrium with gas phase precursors in the study (Nenes, 1998). The secondary organic
aerosol (SOA) is calculated using the volatility basis-set (VBS) modeling method, with
contributions from glyoxal and methylglyoxal. Detailed information can be found in Li et al.
(2010, 2011b).
A persistent air pollution episode from 05 December 2015 to 04 January 2016 in NCP is
simulated using the WRF-CHEM model. Figure 1a shows the model simulation domain, and
detailed model configurations can be found in Table 2.
**2.2   Aerosol radiative module**



123  In the present study, Goddard shortwave module developed by Chou and Suarez (1999,

124 2001) is employed to take into account the ARF effect on the haze formation. The aerosol

125 radiative module developed by Li et al. (2011b) has been incorporated into the WRF-CHEM

126 model to calculate the aerosol optical depth (AOD or $\tau_a$), single scattering albedo (SSA or

127 $\omega_a$), and the asymmetry factor ($g_a$).

128  In the CMAQ aerosol module, aerosols are represented by a three-moment approach

129 with a lognormal size distribution:

130   $n(lnD) = \frac{N}{\sqrt{2\pi}ln\sigma_g}exp[-\frac{1}{2}(\frac{lnD-lnD_g}{ln\sigma_g})^2]$     (1)

131 Where D is the particle diameter, N is the number distribution of all particles in the

132 distribution, $D_g$ is the geometric mean diameter, and $\sigma_g$ is the geometric standard deviation.

133 To calculate the aerosol optical properties, the aerosol spectrum is first divided into 48 bins

134 from 0.002 to 2.5 μm, with radius $r_i$. The aerosols are classified into four types: (1)

135 internally mixed sulfate, nitrate, ammonium, hydrophilic organics and black carbon, and

136 water; (2) hydrophobic organics; (3) hydrophobic black carbon; and (4) other unidentified

137 aerosols. These four kinds of aerosols are assumed to be mixed externally. For the internally

138 mixed aerosols, the complex refractive index at a certain wavelength ($\lambda$) is calculated based

139 on the volume-weighted average of the individual refractive index. Given the particle size

140 and complex refractive index, the extinction efficiency ($Q_e$), $\omega_a$ and $g_a$ are calculated

141 using the Mie theory at a certain wavelength ($\lambda$). The look-up tables of $Q_e$, $\omega_a$ and $g_a$ are

142 established according to particle sizes and refractive indices to avoid multiple Mie scattering

143 calculation. The aerosol optical parameters are interpolated linearly from the look-up tables

144 with the calculated refractive index and particle size in the module.

145  The aerosol optical depth (AOD or $\tau_a$) at a certain wavelength ($\lambda$) in a given

146 atmospheric layer $k$ is determined by the summation over all types of aerosols and all bins:

147   $\tau_a(\lambda, k) = \sum_{i=1}^{48}\sum_{j=1}^{4} Q_e(\lambda, r_i, j, k)\pi r_i^2 n(r_i, j, k)\Delta Z_k$    (2)





148 where $n(r_i, j, k)$ is the number concentration of $j$-th kind of aerosols in the $i$-th bin. $\Delta Z_k$ is

149 the depth of an atmospheric layer. The weighted-mean values of $\sigma$ and $g$ are then

150 calculated by (d'Almeida et al., 1991):

151 $$\omega_a(\lambda, k) = \frac{\sum_{i=1}^{48}\sum_{j=1}^{4} Q_e(\lambda, r_i, j, k)\pi r_i^2 n(r_i, j, k)\omega_a(r_i, j, k)\Delta Z_k}{\sum_{i=1}^{48}\sum_{j=1}^{4} Q_e(\lambda, r_i, j, k)\pi r_i^2 n(r_i, j, k)\Delta Z_k} \tag{3}$$

152 $$g_a(\lambda, k) = \frac{\sum_{i=1}^{48}\sum_{j=1}^{4} Q_e(\lambda, r_i, j, k)\pi r_i^2 n(r_i, j, k)\omega_a(r_i, j, k)g_a(\lambda, r_i, j, k)\Delta Z_k}{\sum_{i=1}^{48}\sum_{j=1}^{4} Q_e(\lambda, r_i, j, k)\pi r_i^2 n(r_i, j, k)\omega_a(r_i, j, k)\Delta Z_k} \tag{4}$$

153 When the wavelength-dependent $\tau_a$, $\omega_a$, and $g_a$ are calculated, they can be used in the

154 Goddard shortwave module to evaluate the ARF. Detailed information can be found in Li et

155 al. (2011b).

156 **2.3  Data and statistical methods for comparisons**

157  The model performance is validated using the available measurements in NCP, including

158 AOD, SSA, PBLH, downward shortwave flux (SWDOWN), aerosol species, and air

159 pollutants. The daily AOD is retrieved from Terra- and Aqua- Moderate Resolution Imaging

160 Spectroradiometer (MODIS) level 2 products, with a resolution of 0.1°×0.1°. The hourly SSA

161 is calculated using the measurement of the turbidity meter at the National Center for

162 Nanoscience and Technology (NCNST), Chinese Academy of Sciences (116.33°E, 39.99°N)

163 in Beijing (Figure 1b). The daily PBLH at 12:00 Beijing time (BJT) is diagnosed from the

164 radiosonde observation at a meteorological site (116.47°E, 39.81°N) in Beijing. The

165 SWDOWN is measured by CM-11 pyranometers at four sites from Chinese Ecosystem

166 Research Network (CERN) in NCP (Liu et al., 2016). The hourly measurements of $O_3$, $NO_2$,

167 $SO_2$, CO and $PM_{2.5}$ concentrations have been released by the China's Ministry of Ecology

168 and Environment (China MEP) since 2013. The hourly submicron sulfate, nitrate, ammonium,

169 and organic aerosols are measured by the Aerodyne Aerosol Chemical Speciation Monitor

170 (ACSM) at NCNST. The primary organic aerosol (POA) and SOA concentrations are

171 obtained from the ACSM measurement analyzed using the Positive Matrix Factorization



(PMF). In addition, we have also analyzed the relationship between near-surface [PM$_{2.5}$] and
the PBLH retrieved from the Lidar measurement at the Institute of Remote Sensing and
Digital Earth (IRSDE), Chinese Academy of Sciences (116.38°E, 40.00°N) in Beijing (Figure
1b).
In the present study, the mean bias (*MB*), root mean square error (*RMSE*) and the index
of agreement (*IOA*) are used to assess the performance of WRF-CHEM model simulations
against measurements. *IOA* describes the relative difference between the model and
observation, ranging from 0 to 1, with 1 indicating perfect agreement.
$$MB = \frac{1}{N}\sum_{i=1}^{N}(P_i - O_i) \tag{5}$$
$$RMSE = \left[\frac{1}{N}\sum_{i=1}^{N}(P_i - O_i)^2\right]^{\frac{1}{2}} \tag{6}$$
$$IOA = 1 - \frac{\sum_{i=1}^{N}(P_i - O_i)^2}{\sum_{i=1}^{N}(|P_i - \overline{O}| + |O_i - \overline{O}|)^2} \tag{7}$$
Where $P_i$ and $O_i$ are the predicted and observed value of a variable, respectively. *N* is the
total number of the predictions used for comparisons, and $\overline{P}$ and $\overline{O}$ represents the average
of the prediction and observation, respectively.

**3    Results and discussions**
**3.1  Model performance**
We first define the base simulation in which the ARF is considered (hereafter referred to
as $f_{base}$), and results from $f_{base}$ are compared to observations in NCP.
**3.1.1 Air pollutants simulations in NCP**
Figure 2 shows the spatial pattern of calculated and observed average near-surface
concentrations of PM$_{2.5}$, O$_3$, NO$_2$, and SO$_2$ along with simulated winds from 05 December
2015 to 04 January 2016 in Eastern China. In general, the simulated air pollutants
distributions are in good agreement with the measurements, but model biases still exist. The



simulated winds are weak or calm during the simulation period, facilitating accumulation of
air pollutants and causing the serious air pollution in Eastern China. NCP is the most polluted
region due to its massive air pollutants emissions, with the average near-surface $[PM_{2.5}]$
generally exceeding 115 µg m$^{-3}$. The highest average near-surface $[PM_{2.5}]$ of more than 150
µg m$^{-3}$ are observed in Beijing, Hebei, Henan, Shandong, and the Guanzhong basin, which
are well reproduced by the model. The simulated $O_3$ concentrations are rather low in NCP,
ranging from 5 to 40 µg m$^{-3}$, consistent with measurements. The low $O_3$ concentration during
wintertime haze episodes in NCP is primarily caused by the weak insolation further
attenuated by clouds and aerosols, the titration of high $NO_x$ emissions, and lack of the $O_3$
transport from outside (Li et al., 2018). Although significant effort has been made to mitigate
air pollutants emissions in NCP, the observed and simulated average $NO_2$ and $SO_2$
concentrations are still high, varying from 30 to 100 µg m$^{-3}$ and 20 to 100 µg m$^{-3}$,
respectively. Interestingly, the simulated high $SO_2$ concentrations are mainly concentrated in
cities and their surrounding areas, but the uniform distribution of $NO_2$ concentrations is
predicted in NCP, showing the substantial contribution of area sources.
Figure 3 shows the temporal profiles of observed and calculated near-surface $PM_{2.5}$, $O_3$,
$NO_2$, $SO_2$ and CO concentrations averaged over monitoring sites in NCP from 05 December
2015 to 04 January 2016. The model generally tracks well the diurnal variation of
near-surface $[PM_{2.5}]$ in NCP, with *IOA* of 0.94, but slightly overestimates $[PM_{2.5}]$, with a *MB*
of 8.3 µg m$^{-3}$. The model successfully reproduces the temporal variations of near-surface $O_3$
concentrations compared to observations in NCP, e.g., peak $O_3$ concentrations in the
afternoon due to active photochemistry and low $O_3$ concentrations during nighttime caused
by the $NO_x$ titration, with an *IOA* of 0.94. However, the model generally underestimates the
$O_3$ concentration during nighttime, with a *MB* of -3.6 µg m$^{-3}$. The model also reasonably well
yields the $NO_2$ diurnal profiles with peaks in the evening, with an *IOA* of 0.86 and a *MB* of



1.6 µg m$^{-3}$, but sometimes there are considerable overestimations and underestimations. The
model generally performs reasonably in predicting the temporal variation of SO$_2$
concentrations against measurements, with an *IOA* of 0.74. However, considering that SO$_2$ is
mainly emitted from point sources and its simulations are more sensitive to the wind field
uncertainties (Bei et al., 2017), the overestimation and underestimation for the SO$_2$
simulation is rather large, with a *RMSE* of 13.3 µg m$^{-3}$. Compared with measurements, the
temporal profile of the near-surface CO concentration in NCP is well simulated, with the *IOA*
and *MB* of 0.87 and 0.1 µg m$^{-3}$, respectively.
**3.1.2 Aerosol species simulations in Beijing**
Figure 4 provides the temporal variations of simulated and observed aerosol species at
NCNST in Beijing from 05 December 2015 to 04 January 2016. Generally, the WRF-CHEM
model predicts reasonably the temporal variations of the aerosol species against the
measurements. The WRF-CHEM model yields the main peaks of the POA concentration
compared to observations in Beijing, but frequently underestimates or overestimates the POA
concentration, with an *IOA* of 0.80 and a *RMSE* of 17.4 µg m$^{-3}$. The POA level in Beijing is
influenced by local emissions and to a large extent trans-boundary transport from outside
during haze days, so its simulation is sensitive to uncertainties from emissions and
meteorological fields (Bei et al., 2010, 2012). The model still has difficulties in simulating
the SOA concentrations, although the VBS modeling method is used and contributions from
glyoxal and methylglyoxal are included in the study, with *IOA* and *MB* of 0.77 and -10.6 µg
m$^{-3}$, respectively. Except the SOA formation and transformation mechanism in the
atmosphere, which remains elusive, many factors have potentials to influence the SOA
simulation, such as meteorology, measurements, precursors emissions, and SOA treatments
(Li et al., 2011a). The model reasonably tracks the temporal variation of the observed sulfate
concentration, and the *MB* and *IOA* are 0.6 µg m$^{-3}$ and 0.90, respectively. Aside from SO$_2$





emissions and simulated meteorological fields, the $SO_2$ oxidation mechanism in the
atmosphere also plays an important role in the sulfate simulation. In addition to direct
emissions and $SO_2$ gas-phase oxidations by hydroxyl radicals (OH) and stabilized criegee
intermediates (sCI), the $SO_2$ oxidation in aerosol water by $O_2$ catalyzed by $Fe^{3+}$ is considered
(Li et al., 2017a). Recent studies have proposed that the aqueous oxidation of $SO_2$ by $NO_2$
under the condition of high RH and $NH_3$ neutralization could interpret the efficient sulfate
formation during wintertime haze events (Wang et al., 2016; Cheng et al., 2016). However,
the mechanism is still not included in this study, which might further improve the sulfate
simulation. The model performs well in simulating the nitrate and ammonium concentrations
against observations in Beijing, with *IOAs* of 0.90 and 0.91, respectively.
**3.1.3 Aerosol radiative properties simulations in NCP**
Aerosol radiative forcing mainly depends on AOD, SSA, and asymmetry parameter (***g***).
The model validations of AOD and SSA are provided in this study to further evaluate the
aerosol radiative effect on the air pollution. The daily AOD at 550 nm, retrieved from Terra-
and Aqua- MODIS level 2 products, is compared with the simulation. Figure 5a shows the
scatter plot of the daily retrieved and simulated AOD averaged in NCP from 05 December
2015 to 04 January 2016. The simulated daily average AOD correlates well with the
observation, with a correlation coefficient of 0.86. Generally, the retrieved and simulated
AOD increases with deterioration of the haze pollution, but the model considerably
underestimates the AOD against the observation. Figure 5b presents the Taylor diagram
(Taylor, 2001) to show the variance, bias and correlation of the simulated and retrieved AOD
from 05 December 2015 to 04 January 2016. There exists a good relationship between the
simulated and retrieved daily AOD during the study episode, with correlation coefficients
generally ranging from 0.5 to 0.9, and standard deviation mostly varying from 0.25 to 1.0.
Figure 6 shows the pattern comparison of the retrieved and simulated AOD averaged during





the simulation period. The model reasonably reproduces the AOD distribution compared to
the observations in NCP, but considerably underestimates the AOD. The simulated and
retrieved AOD averaged in NCP during the simulation period is 0.43 and 0.59, respectively. It
is worth noting that the simulated AOD is not only dependent on the column aerosol content
and constituent, but is also significantly influenced by the relative humidity (RH) controlling
the aerosol hydroscopic growth. Additionally, the satellite retrieved AOD is subject to
contamination by existence of clouds, and considering the high occurrence frequency of
clouds during haze days, the retrieved AOD is generally higher than the simulation
(Engstrom and Ekman, 2010; Chand et al., 2012; Grandey et al., 2013).

Aerosols are the mixture of absorbing and scattering constituents in the atmosphere.

Their radiative effect of cooling or warming the atmosphere relies on many parameters, and
SSA is one of the most important (Satheesh et al., 2010). Figure 7 depicts the comparison of
the measured and simulated diurnal profiles of SSA at NCNST in Beijing during the episodes.
The model performs reasonably in simulating the daily variation of SSA in Beijing, with an
*IOA* of 0.69 and a *MB* of 0.0, but the overestimation or underestimation is rather large. SSA is
the ratio of scattering to extinction, which is highly sensitive to the relative distribution of
scattering and absorbing aerosol constituents in the atmosphere, and the RH determining the
hygroscopic growth of aerosols. Therefore, the uncertainties of the simulated SSA probably
originated from the model biases of aerosol constituents and the RH.
**3.1.4 Downward solar radiation simulations in North China Plain**

Figure 8 presents the daily profiles of simulated and observed SWDOWN at ground

surfaces in Beijing, Jiaozhouwan, Luancheng, and Yuancheng from 05 December 2015 to 04
January 2016. The WRF-CHEM model simulates well the daily variation of SWDOWN,
especially in Jiaozhouwan, Luancheng, and Yucheng, with *IOAs* around 0.90. The model is
subject to overestimating the SWDOWN against measurements, with *MB*s ranging from 6.3



to 86.2 W m$^{-2}$. The SWDOWN reaching the ground surface is very sensitive to the cloud cover and optical thickness. However, the WRF-CHEM model still has difficulties in accurately predicting the cloud cover and optical thickness, which might constitute one of the most important reasons for model biases of the SWDOWN. In addition, the horizontal resolution used in simulations cannot adequately resolve the cumulus clouds, also causing uncertainties in the simulations of the SWDOWN.

**3.1.5 PBLH simulations in Beijing**

Figure 9 shows the temporal variations of the observed and simulated PBLH at a meteorological site in Beijing from 05 December 2015 to 04 January 2016. The average PBLH at 12:00 BJT during the episode at the meteorological site is 465.2 m, with the minimum of 101.8 m and the maximum of 1017.9 m, showing decreased PBLH during the haze episode. In general, the WRF-CHEM model tracks reasonably the daily variation of the PBLH in Beijing, with an *IOA* of 0.70. However, the model has difficulties in reproducing the observed very low PBLH, e.g., less than 200 m. The PBLH varies substantially with time due to many factors including large-scale dynamics, cloudiness, convective mixing, and the diurnal cycle of solar radiation (Sivaraman et al., 2013). Therefore, the simulation uncertainties of meteorological conditions constitute the main reason for the simulation bias of PBLH. For example, the overestimation of SWDOWN at 12:00 BJT (Figure 8a) probably caused the overestimation of PBLH in Beijing.

In general, the simulated variations of SWDOWN, PBLH, aerosol radiative properties, air pollutants (PM$_{2.5}$, O$_3$, NO$_2$, SO$_2$, CO) and aerosol species are in good agreement with observations, indicating that the simulations of meteorological conditions, chemical processes and the emission inventory used in the WRF-CHEM model are reasonable, providing a reliable basis for the further investigation.

**3.2 Relationship between near-surface [PM$_{2.5}$] and PBLH**



Figure 10 presents the scatter plot of the Lidar retrieved PBLH at IRSDE and
near-surface [PM$_{2.5}$] at a monitoring site close to IRSDE during daytime (08:00 ~ 17:00 LT)
from 08 January to 20 February 2014. The wind speeds (WSPD) at a meteorological site
close to IRSDE are shown by the color of the filled circles in Figure 10. Additionally,
near-surface [PM$_{2.5}$] during daytime are also subdivided into 20 bins with the interval of 25
μg m$^{-3}$. The PBLH as the bin of near-surface [PM$_{2.5}$] is assembled, and an average of PBLH
in each bin is calculated (Nakajima et al., 2001; Kawamoto et al., 2006), which is represented
by the rectangle in Figure 10. Generally, on average, when the PBLH decreases from 1500 m
to around 400 m, the near-surface [PM$_{2.5}$] increase from 10 to more than 200 μg m$^{-3}$. When
near-surface [PM$_{2.5}$] exceed 200 μg m$^{-3}$, the PBLH remains 400~500 m. Previous studies
have also reported the nonlinear relationship between the PBLH and near-surface [PM$_{2.5}$],
and proposed that increasing [PM$_{2.5}$] reduce the PBLH or the ARF is attributed to the PBLH
decrease (e.g., Petaja et al., 2016; Tie et al., 2017; Liu et al., 2018).
The PBLH is primarily determined by the wind shear in the vertical direction and the
thermal condition of ground surfaces. The occurrence of low near-surface [PM$_{2.5}$] generally
corresponds to efficient dispersions of PM$_{2.5}$ in horizontal and/or vertical directions. The
strong horizontal winds in the lower atmosphere not only disperse PM$_{2.5}$ emitted or formed
efficiently, but also intensify the wind shear in the vertical direction, increasing the PBLH
and facilitating the rapid vertical exchange of PM$_{2.5}$ in the PBL. When near-surface [PM$_{2.5}$]
are less than 50 μg m$^{-3}$, the PBLH exceeding 1000 m is observed, which is chiefly
determined by strong horizontal winds and less influenced by the ground thermal condition
during wintertime, and the observed average WSPD is about 2.4 m s$^{-1}$. The occurrence of
high near-surface [PM$_{2.5}$] indicates that the lower atmosphere is stable or stagnant, with weak
horizontal winds and inactive convections, hindering the dispersion of PM$_{2.5}$ in the horizontal
and vertical directions. Additionally, as the horizontal winds become weak or calm, the wind



shear in the vertical direction is diminished and the PBLH is dominated by the ground
thermal condition. When near-surface [$PM_{2.5}$] increase from 50 to around 200 µg m$^{-3}$, the
PBLH decreases from around 700 to 400 m, and the average WSPD decreases to 1.8 m s$^{-1}$.
However, the increased $PM_{2.5}$ reducing PBLH still cannot be fully attributed to the ARF,
which is more likely caused by the decrease of winds or the formation of stagnant situations
in the low-level atmosphere. When near-surface [$PM_{2.5}$] exceed 200 µg m$^{-3}$, the observed
PBLH fluctuates between 400 and 500 m with the average WSPD of around 1.0 m s$^{-1}$, and
does not exhibit continuous decrease with the increasing near-surface [$PM_{2.5}$].

Under the stagnant situation with weak winds, the PBLH is more sensitive to the ground

thermal condition. Increasing aerosols or $PM_{2.5}$ in the low-level atmosphere attenuate the
SWDOWN to the ground surface and decrease the surface temperature (TSFC) and
turbulence kinetic energy, suppressing the PBL development and further enhancing
near-surface [$PM_{2.5}$]. Therefore, with near-surface [$PM_{2.5}$] exceeding 200 µg m$^{-3}$, the inert
PBLH might be caused by the defect of the Lidar retrieved PBLH. The aerosol backscatter
signal received by Lidar is used to retrieve the PBLH. If the atmosphere is stable, the aerosols
near the maximal PBLH are subject to being confined in situ, and the retrieved PBLH is
generally the maximal one. Additionally, it is worth noting that the occurrence of the
wintertime severe haze pollution in NCP is often accompanied with the high-level
convergence between 500 and 700 hPa, producing a persistent and strong sinking motion in
the mid-lower troposphere to reduce the PBLH and facilitate accumulation of air pollutants
(Wu et al. 2017; Ding et al., 2017). Therefore, a subsidence inversion appears in the lower
layer as a result of the air masses sinking in the middle-troposphere, restraining the PBL
development and determining the maximal PBLH. Hence, it is imperative to evaluate the
ARF to the PBLH and near-surface [$PM_{2.5}$].
**3.3  Sensitivity studies**





The conceptual model about the ARF contribution to the heavy haze formation has been
established in previous studies (e.g., Tie et al., 2017; Liu et al., 2018). During wintertime,
under stagnant meteorological situations with weak winds and humid air, air pollutants are
subject to accumulation in the PBL, facilitating the formation of $PM_{2.5}$. Increasing $PM_{2.5}$ in
the PBL absorbs or scatters the incoming solar radiation to decrease the TSFC and facilitate
anomalous temperature inversion, subsequently suppressing the vertical turbulent diffusion
and decreasing the PBLH to further trap more air pollutants and water vapor to increase the
RH in the PBL. Increasing RH enhances aerosol hygroscopic growth and multiphase
reactions and augments the particle size and mass, causing further dimming and decrease of
the TSFC and PBLH. The whole process constitutes a positive feedback induced by the
aerosol radiation effect to enhance near-surface $[PM_{2.5}]$, which has been proposed in many
studies (Quan et al., 2013; Petaja et al., 2016; Yang et al., 2016; Tie et al., 2017; Ding et al.,
2017; Liu et al., 2018). The noted positive meteorological condition feedback has also been
considered as the main reason for the near-surface $PM_{2.5}$ explosive growth (Zhong et al., 2018;
X. Y. Zhang et al., 2018).
To comprehensively evaluate the influence of the ARF on near-surface $[PM_{2.5}]$ during
the haze episode, a sensitivity study has been conducted, in which the ARF is turned off
(hereafter referred as $f_{rad0}$). Therefore, the contribution of the ARF to near-surface $[PM_{2.5}]$
can be determined by the difference between $f_{base}$ and $f_{rad0}$ ($f_{base}$- $f_{rad0}$). The most
polluted area in NCP is first selected to verify the conceptual model of the ARF contribution
to the heavy haze formation, with the average near-surface $[PM_{2.5}]$ during the haze episode
exceeding 150 μg m$^{-3}$. Figure 11 provides the temporal variation of near-surface $[PM_{2.5}]$,
SWDOWN, TSFC, PBLH, and RH averaged in the selected area during the episode in $f_{base}$
and $f_{rad0}$. Apparently, the ARF considerably decreases the solar radiation reaching the
ground surface and correspondingly lowers the TSFC (Figures 11b and 11c). Subsequently,





the PBLH is decreased and the surface RH is increased due to decreasing TSFC during
daytime (Figures 11d and 11e). However, the variation trend of near-surface [PM$_{2.5}$], PBLH,
TSFC and RH due to the ARF is not similar to that proposed in the conceptual model. During
the haze development stage, whether the ARF is considered or not, the TSFC and RH exhibit
an increasing trend, showing the air mass originated from the south, and the PBLH does not
consistently   decreases   with   increasing   near-surface   [PM$_{2.5}$].   Additionally,   the   ARF
contribution to near-surface [PM$_{2.5}$] is generally marginal during the haze development stage.
During the haze maturation stage, the ARF commences to elevate near-surface [PM$_{2.5}$]
appreciably. It is worth noting that, even if the ARF is not considered in $f_{rad0}$, the heavy
haze pollution still occurs during the episode. For example, from 17 to 20 December 2015,
without the ARF, near-surface [PM$_{2.5}$] still continue to increase from around 30 to 300 μg m$^{-3}$,
and fluctuates between 150 to 300 μg m$^{-3}$ until the occurrence of favorable meteorological
conditions on 25 December. Hence, according to the variation trend of near-surface [PM$_{2.5}$]
with and without the ARF contribution, the continuous accumulation of PM$_{2.5}$ during the haze
episode is not primarily caused by the ARF, but predominantly induced by the stagnant
meteorological conditions as well as the massive air pollutants emissions in NCP.

In order to quantitatively evaluate effects of the ARF on near-surface [PM$_{2.5}$], which

cannot be reflected by the temporal variation of near-surface [PM$_{2.5}$], TSFC, PBLH and RH,
an ensemble method is used in this study. The daytime near-surface [PM$_{2.5}$] in NCP during
the episode in $f_{base}$ are first subdivided into 30 bins with an interval of 20 μg m$^{-3}$. The
SWDOWN, TSFC, PBLH, the near-surface WSPD, RH, and [PM$_{2.5}$] in $f_{base}$ and $f_{rad0}$ in
the same grid cell are assembled as the bin [PM$_{2.5}$], respectively, and an average of these
variables in each bin are calculated. Figure 12 shows the decrease of SWDOWN (%), TSFC
($^{o}$C), PBLH (%), WSPD (m s$^{-2}$), and the increase of RH (%, not percentage change) and
near-surface [PM$_{2.5}$] contribution (%) caused by the ARF as a function of bin [PM$_{2.5}$]. The



SWDOWN reaching the ground surface almost decreases linearly with the enhancement of
near-surface [PM$_{2.5}$]. When the ARF is considered, aerosols in the atmosphere absorb or
scatter the incoming solar radiation, directly attenuating the radiation reaching the ground
surface. When near-surface [PM$_{2.5}$] exceed 200 μg m$^{-3}$, the SWDOWN at ground surfaces
decreases by more than 20% (Figure 12a). Moreover, the decrease of the SWDOWN
correspondingly lowers the TSFC and the decrease of the TSFC is generally proportional to
near-surface [PM$_{2.5}$], about 0.35 °C per 100 μg m$^{-3}$ PM$_{2.5}$ (Figure 12b). Interestingly, the ARF
also decreases near-surface WSPD by about 0.1~0.2 m s$^{-1}$ with near-surface [PM$_{2.5}$]
exceeding 80 μg m$^{-3}$ (Figure 12c). When severe air pollution occurs in NCP during
wintertime, atmospheric convergence occurs in the PBL (Liao et al., 2015; Ding et al., 2017).
However, the ARF induced cooling in the low-level air generates a divergence in NCP,
causing the decrease of near-surface WSPD.
The PBLH is primarily determined by the atmospheric dynamic and thermal condition of
ground surfaces. Therefore, the decrease of WSPD and TSFC due to the ARF subsequently
suppresses the PBL development and diminishes the PBLH (Figure 12d). When near-surface
[PM$_{2.5}$] are less than 250 μg m$^{-3}$, the PBLH decreases rapidly with increasing [PM$_{2.5}$]. When
the near-surface [PM$_{2.5}$] are between 250 μg m$^{-3}$ and 350 μg m$^{-3}$, the decrease of PBLH is
around 28%. With near-surface [PM$_{2.5}$] more than 350 μg m$^{-3}$, the decrease of PBLH exceeds
30%. As for the ARF effect on water vapor in the PBL, the conceptual model has proposed
that the decreased PBL induced by the ARF weakens the vertical exchange of water vapor or
the dispersion of water vapor is constrained by the shallow PBL (Tie et al., 2017; Liu et al.,
2018). However, Figure 13a shows that the ARF decrease the near-surface water vapor
content slightly, by more than 0.1 g kg$^{-1}$ with near-surface [PM$_{2.5}$] exceeding 100 μg m$^{-3}$.
During the haze episode in NCP, the abundant moisture in the PBL is mainly transported
from the south. The divergence due to cooling caused by the ARF weakens the prevailing



southerly wind and decreases the moisture transport from the south, reducing the water vapor
content in NCP. Considering that the RH is sensitive to the temperature with a constant water
vapor content, the ARF induced cooling still increases the near-surface RH (Figure 12e).
When near-surface $[PM_{2.5}]$ exceed 300 μg m$^{-3}$, the RH is increased by more than 5%, so the
heavy haze generally causes the air to be more humid.

More $PM_{2.5}$ emitted or formed are trapped by a shallow PBL caused by the ARF, and

increased RH promotes the aerosol hygroscopic growth and further multiphase reactions,
progressively enhancing near-surface $[PM_{2.5}]$ (Figure 12f). When near-surface $[PM_{2.5}]$ are
more than 50 μg m$^{-3}$, the contribution of the ARF to near-surface $[PM_{2.5}]$ consistently
increases with the haze deterioration. When the severe haze occurs, i.e., near-surface $[PM_{2.5}]$
exceed 250 μg m$^{-3}$, more than 12% or 30 μg m$^{-3}$ $PM_{2.5}$ is contributed by the ARF. The
simulated ARF effects on near-surface $[PM_{2.5}]$ are generally comparable to those reported by
previous studies. Z. Wang et al. (2014) have shown that the ARF increases the monthly $PM_{2.5}$
concentration by 10%-30% in Beijing-Tianjin-Hebei in January 2013. Using the
WRF-CHEM model, Gao et al. (2015) have indicated that the ARF increases the $PM_{2.5}$
concentration by 10-50 μg m$^{-3}$ (2%-30%) over Beijing, Tianjin, and south Hebei from 10 to
15 January 2013, a period with the simulated maximum hourly surface $PM_{2.5}$ concentration
of more than 600 μg m$^{-3}$. X. Zhang et al. (2018) have also quantified the aerosol-meteorology
interaction effect on $PM_{2.5}$ concentrations in China in 2014 using the WRF-CHEM model,
showing that the increase of $PM_{2.5}$ concentrations associated with the ARF is up to 16% in
China. Other previous studies have also confirmed the ARF effect during the heavy haze
pollution episode (Wang et al., 2015; Zhang et al., 2015; Gao et al., 2016). However, when
near-surface $[PM_{2.5}]$ are less than 50 μg m$^{-3}$, the contribution of the ARF to near-surface
$[PM_{2.5}]$ is negative, although the ARF decreases PBLH and increases RH. One of the possible
reasons for the negative contribution of the ARF is perturbations of wind fields caused by the



ARF induced cooling. Figure 13b presents the average vertical velocity below about 400 m in
$f_{rad0}$ as a function of near-surface [$PM_{2.5}$]. Apparently, when the ARF is not considered, the
area with near-surface [$PM_{2.5}$] less than 100 μg m$^{-3}$ is generally controlled by downward
airflow, and vice versa for the area with near-surface [$PM_{2.5}$] more than 100 μg m$^{-3}$. The ARF
induced cooling generally cause a downward motion in the PBL (Figure 13c), which
suppresses the upward motion in the area with near-surface [$PM_{2.5}$] more than 100 μg m$^{-3}$ to
enhance near-surface [$PM_{2.5}$], but accelerates the downward motion in the area with
near-surface [$PM_{2.5}$] less than 100 μg m$^{-3}$ to decrease near-surface [$PM_{2.5}$]. Countered by the
decrease of PBLH and increase of RH, the ARF contribution becomes positive with
near-surface [$PM_{2.5}$] exceeding 50 μg m$^{-3}$.
Figure 14 presents spatial distributions of the average near-surface $PM_{2.5}$ contribution
due to the ARF during the episode. The average near-surface $PM_{2.5}$ contribution caused by
the ARF in NCP is 10.2 μg m$^{-3}$ or 7.8%, with the maximum exceeding 40 μg m$^{-3}$ in the south
of Hebei. On average, the ARF contribution to near-surface [$PM_{2.5}$] is the most significant in
Tianjin, about 17.6 μg m$^{-3}$ or 10.3%, followed by Hebei (11.6 μg m$^{-3}$ or 9.3%), Shandong
(11.5 μg m$^{-3}$ or 7.3%), Henan (11.2 μg m$^{-3}$ or 7.7%), Anhui (7.7 μg m$^{-3}$ or 7.4%), Beijing (7.3
μg m$^{-3}$ or 6.9%), and Jiangsu (7.0 μg m$^{-3}$ or 6.2%). It is noteworthy that the ARF contribution
during the episode in North China is generally positive, but in its surrounding area the
contribution becomes negative. At a large scale, when the air pollution occurs during
wintertime in North China, the vertical motion over the polluted area generally shows an
ascending-descending-ascending distribution from the surface to the middle level of the
troposphere, and wind directions present a structure of convergence-divergence-convergence
accordingly (Liao et al., 2015; Wu et al., 2017; Ding et al., 2017). The ARF cools the
low-level atmosphere and induces a downward motion, which suppresses the upward motion
in the convergence area in North China to increase near-surface [$PM_{2.5}$], but accelerates the



downward motion in the divergence area to decrease [PM$_{2.5}$].

Furthermore, when the ARF is considered, near-surface [PM$_{2.5}$] over the East and South

China Sea are also increased, with an enhancement less than 5 µg m$^{-3}$ (about 3% to more than
15%). Considering the low near-surface [PM$_{2.5}$] over sea, the [PM$_{2.5}$] enhancement might be
caused by the PM$_{2.5}$ transport from the continent. Figure 15 shows the spatial distribution of
the TSFC and wind field variation caused by the ARF averaged during the episode.
Apparently, the ARF causes a widespread cooling effect in East China, and the cooling is the
most significant in NCP, with the maximum TSFC decrease exceeding 1.5$^{o}$C. The cooling
effect in NCP induces a weak northerly wind, decreasing the prevailing southerly wind
during the haze episode (Figure 15). Additionally, the cooling effect over the continent also
intensifies the temperature contrast between land and sea, producing a secondary circulation
to transport the PM$_{2.5}$ from the continent to the East and South China Sea.

**4    Summary and conclusions**

In the study, a persistent haze pollution episode in NCP from 05 December 2015 to 04

January 2016 are simulated using the WRF-CHEM model to verify the ARF contribution to
the haze formation. Generally, the model reproduces well the spatial distributions and
temporal variations of PM$_{2.5}$, O$_3$, NO$_2$, SO$_2$, and CO mass concentrations against observations
in NCP. The calculated temporal variations of aerosol species are also consistent with the
ACSM measurement in Beijing, particularly with regard to the simulation of sulfate, nitrate,
and ammonium. Moreover, the model simulates reasonably well the variation of SWDOWN,
PBLH, and aerosol radiative properties during the episode, compared to the measurement.

Previous studies have established that a positive feedback induced by the ARF causes

the heavy haze formation by modulating the PBL and RH. However, model results
demonstrate that during the haze development stage in NCP, the ARF does not dominate





accumulation of near-surface [PM$_{2.5}$]. The TSFC and RH generally exhibit an increasing
trend, showing that the air mass originated from the south, and the PBLH does not
consistently decrease as proposed with increasing near-surface [PM$_{2.5}$]. During the haze
maturation stage, the ARF considerably enhances near-surface [PM$_{2.5}$].
Ensemble analyses of model results show that, during daytime, the ARF attenuates
SWDOWN reaching ground surfaces efficiently with increasing near-surface [PM$_{2.5}$] in NCP,
and SWDOWN is decreased by more than 20% when near-surface [PM$_{2.5}$] exceed 200 μg m$^{-3}$.
Correspondingly, the TSFC progressively decreases with increasing near-surface [PM$_{2.5}$],
with a rate of around 0.35 $^{\circ}$C per 100 μg m$^{-3}$ PM$_{2.5}$. The ARF induced cooling generates a
divergence in the low-level atmosphere in NCP, lowering the near-surface WSPD and
decreasing the water vapor transport from the south. The decreased WSPD and TSFC caused
by the ARF hinder the PBL development and the PBLH decreases rapidly with increasing
near-surface [PM$_{2.5}$]. When near-surface [PM$_{2.5}$] exceed 250 μg m$^{-3}$, the PBLH is decreased
by over 28%. Although the water content in NCP is decreased slightly, the RH is still
increased due to the ARF induced cooling, and the RH enhancement exceeds 5% when
near-surface [PM$_{2.5}$] are more than 300 μg m$^{-3}$. A shallow PBL and more humid air caused by
the ARF accelerate the PM$_{2.5}$ accumulation and secondary pollutant formation, facilitating
heavy haze formation. The contribution of the ARF to near-surface [PM$_{2.5}$] increases from 12%
to 20% when near-surface [PM$_{2.5}$] increase from 250 to 500 μg m$^{-3}$. However, the ARF
decreases the PM$_{2.5}$ level with near-surface [PM$_{2.5}$] less than 50 μg m$^{-3}$.
The average near-surface PM$_{2.5}$ contribution of the ARF during the episode in NCP is
10.2 μg m$^{-3}$ or 7.8%. The ARF aggravates the heavy haze formation in North China, but in its
surrounding area the ARF slightly mitigates the haze pollution. Generally, there is a structure
of convergence-divergence-convergence over the polluted area of North China from the
surface to the middle level of the troposphere. The ARF causes a widespread cooling effect in





East China, particularly remarkable in NCP. A downward motion is induced due to the
cooling of the low-level atmosphere, impeding the upward motion in the convergence area in
North China to increase near-surface [$PM_{2.5}$], but accelerating the downward motion in the
divergence area to decrease [$PM_{2.5}$].
Although the model performs generally well in simulating air pollutants, aerosol species
and radiative properties, SWDOWN, and PBLH, the uncertainties from meteorological fields
and emission inventory still have potentials to influence the ARF evaluation. Particularly,
further studies need to be conducted to improve the AOD simulations. In this study, the ARF
only considers the aerosol effect on the solar radiation, and the influence of longwave
radiation also needs to be included. In addition, aerosols play an important role in the cloud
process serving as cloud condensation nuclei (CCN) and ice nuclei (IN). Therefore,
aerosol-cloud interactions (aerosol indirect effect) modify temperature and moisture profiles
and further influence precipitation, leading to potential effects on the atmospheric chemistry
(Wang et al., 2011). Future studies should be performed to investigate the feedbacks of the
aerosol indirect effect on the air pollutants.


*Author contribution.* Guohui Li, as the contact author, provided the ideas and financial
support, developed the model code, verified the conclusions, and revised the paper. Jiarui Wu
conducted a research, designed the experiments, carried the methodology out, performed the
simulation, processed the data, prepared the data visualization, and prepared the manuscript
with contributions from all authors. Naifang Bei provided the treatment of meteorological
data, analyzed the study data, validated the model performance, and reviewed the manuscript.
Bo Hu provided the observation data used in the study, synthesized the observation, and
reviewed the paper. Suixin Liu, Meng Zhou, Qiyuan Wang, Zirui Liu, and Yichen Wang



provided the data and the primary data process, and reviewed the manuscript. Xia Li, Lang
Liu, and Tian Feng analyzed the initial simulation data, visualized the model results and
reviewed the paper. Junji Cao, Xuexi Tie, Jun Wang provided critical reviews pre-publication
stage. Luisa T. Molina provided a critical preview and financial support, and revised the
manuscript.


*Acknowledgements*. This work is financially supported by the National Key R&D Plan
(Quantitative Relationship and Regulation Principle between Regional Oxidation Capacity of
Atmospheric and Air Quality (2017YFC0210000)) and National Research Program for Key
Issues in Air Pollution Control. Luisa Molina acknowledges support from US NSF Award

1560494.










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





Table 1 Impact of the ARF on near-surface [$PM_{2.5}$] in China

| Reference | Time | Location | Impact on [$PM_{2.5}$] |
|---|---|---|---|
| Z. Wang et al. (2014) | January 2013 | Beijing-Tianjin-Hebei | +10~30% |
| J. Wang et al. (2014) | January 2013 | North China Plain | Up to +140 $\mu g\ m^{-3}$ |
| Gao et al. (2015) | 10-15 January 2013 | Beijing, Tianjin, and south Hebei | +10-50 $\mu g\ m^{-3}$ (2-30%) |
| Wang et al. (2015) | 7-11 July 2008 | Beijing, Tianjin, Hebei, East Shanxi, West Shandong, and North Henan | +14% |
| Zhang et al. (2015) | January 2013 | Henan, Hubei, Guangxi, and Sichuan | Maximum +69.3 $\mu g\ m^{-3}$ |
| Ding et al. (2016) | December 2013 | Eastern China and the Sichuan Basin | Up to +100 $\mu g\ m^{-3}$ |
| Gao et al. (2016) | January 2010 | Shijiazhuang | More than +20 $\mu g\ m^{-3}$ |
| X. Y. Zhang et al. (2018) | December 2016 | Beijing | around +84% of [$PM_{2.5}$] during cumulative explosive growth |
| Liu et al. (2018) | 15-21 December 2016 | North China Plain | +56 $\mu g\ m^{-3}$ |
| X. Zhang et al. (2018) | 2014 | China | over +16% for the daily maximum [$PM_{2.5}$] |
| Zhong et al. (2018) | January 2013, February 2014, December 2015, and December 2016 to 10 January 2017 | Beijing | Over +70% of [$PM_{2.5}$] during cumulative explosive growth |




**880**    Table 2 WRF-CHEM model configurations.

**881**

| Region | East Asia |
|---|---|
| Simulation period | 05 December 2015 to 04 January 2016 |
| Domain size | 400 × 400 |
| Domain center | 35°N, 114°E |
| Horizontal resolution | 12 km × 12 km |
| Vertical resolution | 35 vertical levels with a stretched vertical grid with spacing ranging from 30 m near the surface, to 500 m at 2.5 km and 1 km above 14 km |
| Microphysics scheme | WSM 6-class graupel scheme (Hong and Lim, 2006) |
| Cumulus scheme | Grell-Devenyi ensemble scheme (Grell and Devenyi, 2002) |
| Boundary layer scheme | MYJ TKE scheme (Janjić, 2002) |
| Surface layer scheme | MYJ surface scheme (Janjić, 2002) |
| Land-surface scheme | Unified Noah land-surface model (Chen and Dudhia, 2001) |
| Longwave radiation scheme | Goddard longwave scheme (Chou and Suarez, 2001) |
| Shortwave radiation scheme | Goddard shortwave scheme (Chou and Suarez, 1999) |
| Meteorological boundary and initial conditions | NCEP 1°×1° reanalysis data |
| Chemical initial and boundary conditions | MOZART 6-hour output (Horowitz et al., 2003) |
| Anthropogenic emission inventory | Developed by Zhang et al. (2009) and Li et al. (2017), 2012 base year, and SAPRC-99 chemical mechanism |
| Biogenic emission inventory | Online MEGAN model developed by Guenther et al. (2006) |

**882**
**883**
**884**
**885**
**886**





Figure Captions

Figure 1 (a) WRF-CHEM simulation domain with topography and (b) Beijing-Tianjin-Hebei area. In (a), the blue circles represent centers of cities with ambient monitoring sites in, and the size of blue circles denotes the number of ambient monitoring sites of cities. In (b), the blue and red filled circles denote the NCNST and IRSDE site, respectively, the red filled rectangle denotes the meteorological site. The red numbers denote the CERN sites with the solar radiation measurement. 1: Beijing urban; 2: Jiaozhouwan; 3: Yucheng; 4: Luancheng.

Figure 2 Pattern comparisons of simulated (color counters) vs. observed (colored circles) near-surface mass concentrations of (a) $PM_{2.5}$, (b) $O_3$, (c) $NO_2$, and (d) $SO_2$ averaged from 05 December 2015 to 04 January 2016. The black arrows indicate simulated surface winds.

Figure 3 Comparison of observed (black dots) and simulated (solid red lines) diurnal profiles of near-surface hourly mass concentrations of (a) $PM_{2.5}$, (b) $O_3$, (c) $NO_2$, (d) $SO_2$, and (d) CO averaged at monitoring sites in NCP from 05 December 2015 to 04 January 2016.

Figure 4 Comparison of measured (black dots) and simulated (black line) diurnal profiles of submicron aerosol species of (a) POA, (b) SOA, (c) sulfate, (d) nitrate, and (e) ammonium at NCNST site in Beijing from 05 December 2015 to 04 January 2016.

Figure 5 (a) Scatter plot of the MODIS retrieved and simulated daily AOD, (b) Taylor diagram (Taylor, 2001) to present the variance, bias and correlation of the retrieved and simulated daily AOD averaged in NCP from 05 December 2015 to 04 January 2016.

Figure 6 Spatial distribution of (a) retrieved and (b) simulated AOD averaged from 05 December 2015 to 04 January 2016 in NCP.

Figure 7 Comparison of measured (black dots) and predicted (red line) diurnal profiles of SSA in Beijing from 05 December 2015 to 04 January 2016.

Figure 8 Comparison of measured (black dots) and predicted (red line) diurnal profiles of the SWDOWN reaching the ground surface in (a) Beijing, (b) Jiaozhouwan, (c) Luancheng, and (d) Yucheng from 05 December 2015 to 04 January 2016.

Figure 9 Comparison of predicted diurnal profile (red line) of PBLH from 05 December 2015 to 04 January 2016 with observations at 12:00 BJT in Beijing.

Figure 10 Scatter plot of the PBLH and near-surface [$PM_{2.5}$] at IRSDE site from 12 January to 20 February 2014. The black rectangle shows the bin average of PBLH. The color of the filled circles denotes the WSPD at the meteorological site close to IRSDE in Figure 1b.

Figure 11 Temporal variations of the average (a) near-surface [$PM_{2.5}$], (b) SWDOWN at the ground surface, (c) TSFC, (d) PBLH, and (e) RH in the most polluted area in NCP with [$PM_{2.5}$] of more than 150 μg m$^{-3}$ in $f_{base}$ (red solid line) and $f_{rad0}$ (blue solid line) from 05 December 2015 to 04 January 2016.

Figure 12 Average (a) percentage decrease of SWDOWN at the ground surface, (b) decrease of TSFC, (c) decrease of WSPD, (d) percentage decrease of PBLH, (e) increase of RH, and (f) percentage contribution of near-surface [$PM_{2.5}$] caused by the ARF, as a



930 function of the near-surface [PM$_{2.5}$] in NCP during daytime from 05 December 2015
931 to 04 January 2016.

932 Figure 13 Average (a) decrease of water vapor content and (c) increase of average vertical
933 velocity below 400 m caused by the ARF, and (b) average vertical velocity below 400
934 m as a function of the near-surface [PM$_{2.5}$] in NCP during daytime from 05 December
935 2015 to 04 January 2016.

936 Figure 14 Near-surface [PM$_{2.5}$] contribution caused by the ARF, averaged from 05 December
937 2015 to 04 January 2016 in NCP.

938 Figure 15 TSFC and wind filed variations caused by the ARF, averaged from 05 December
939 2015 to 04 January 2016 in NCP.









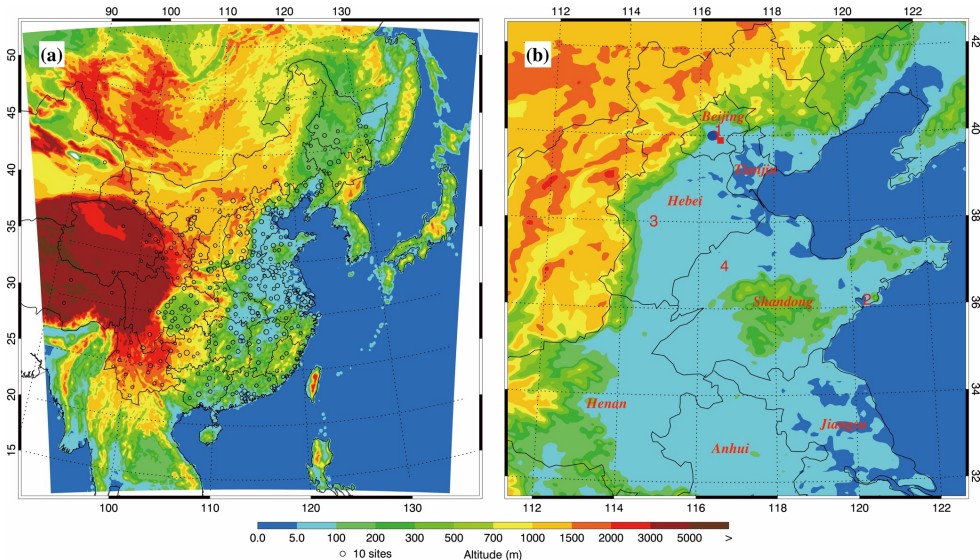

Figure 1 (a) WRF-CHEM simulation domain with topography and (b) North China Plain. In
(a), the blue circles represent centers of cities with ambient monitoring sites in, and the size
of blue circles denotes the number of ambient monitoring sites of cities. In (b), the blue and
red filled circles denote the NCNST and IRSDE site, respectively, and the red filled rectangle
denotes the meteorological site. The red numbers denote the CERN sites with the solar
radiation measurement. 1: Beijing urban; 2: Jiaozhouwan; 3: Yucheng; 4: Luancheng.



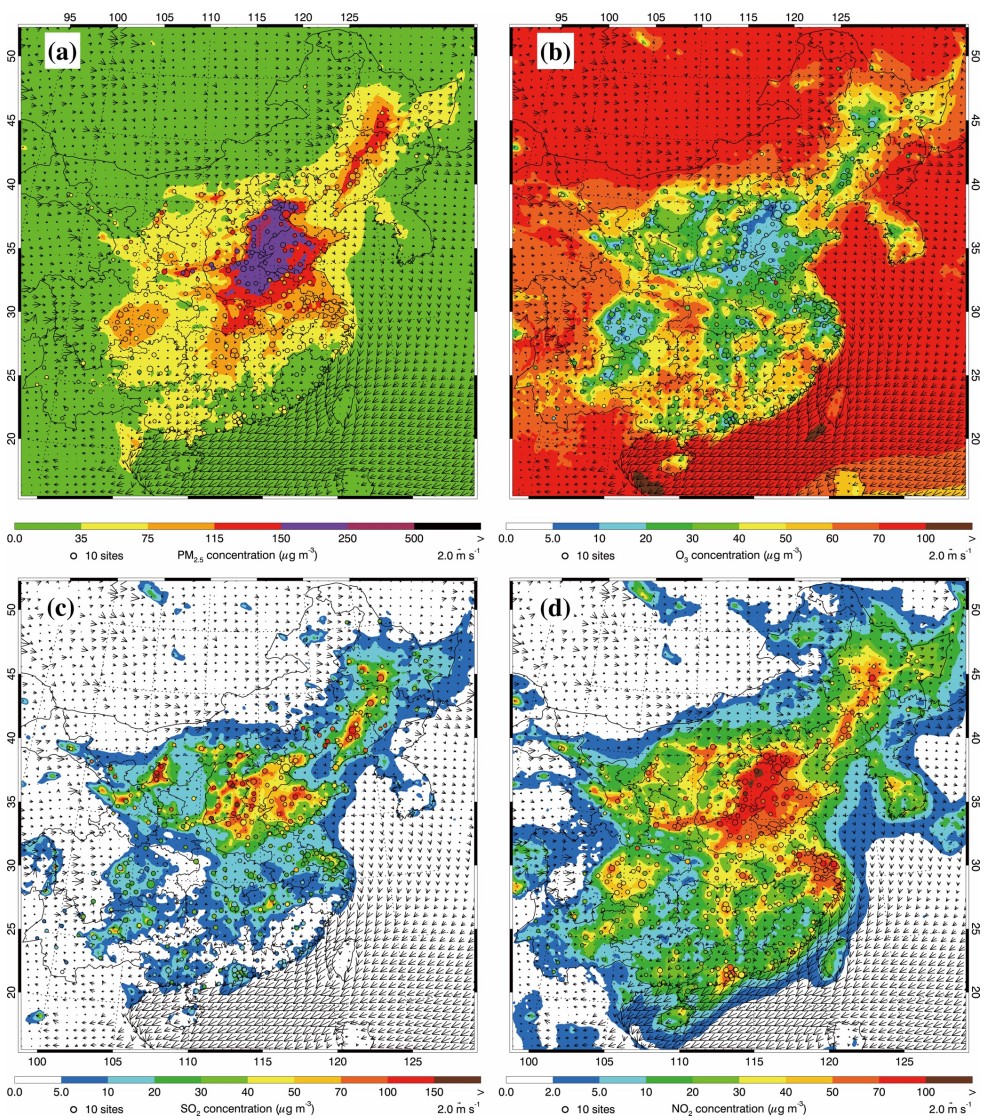

Figure 2 Pattern comparisons of simulated (color counters) vs. observed (colored circles) near-surface mass concentrations of (a) $PM_{2.5}$, (b) $O_3$, (c) $NO_2$, and (d) $SO_2$ averaged from 05 December 2015 to 04 January 2016. The black arrows indicate simulated surface winds.



Figure 3 Comparison of observed (black dots) and simulated (solid red lines) diurnal profiles
of near-surface hourly mass concentrations of (a) PM$_{2.5}$, (b) O$_3$, (c) NO$_2$, (d) SO$_2$, and (d) CO
averaged at monitoring sites in NCP from 05 December 2015 to 04 January 2016.





Figure 4 Comparison of measured (black dots) and simulated (black line) diurnal profiles of
submicron aerosol species of (a) POA, (b) SOA, (c) sulfate, (d) nitrate, and (e) ammonium at
NCNST site in Beijing from 05 December 2015 to 04 January 2016.





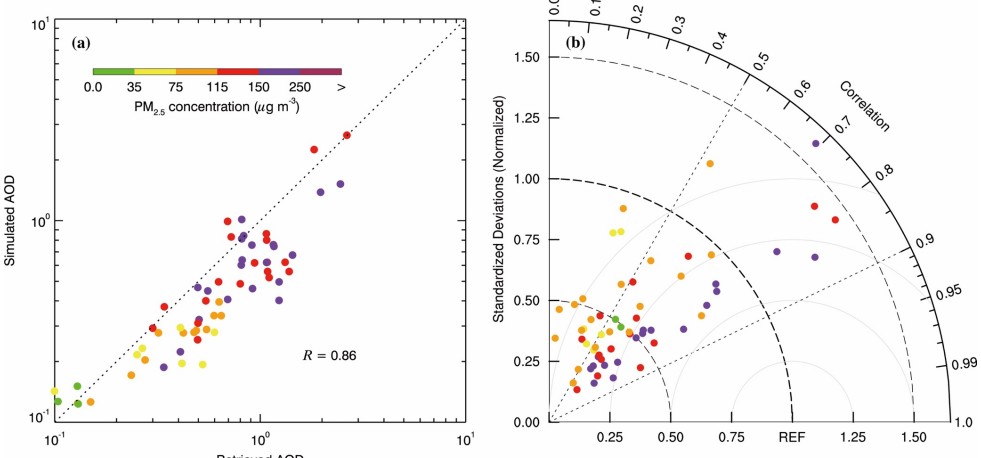

Figure 5 (a) Scatter plot of the MODIS retrieved and simulated daily AOD, (b) Taylor
diagram (Taylor, 2001) to present the variance, bias and correlation of the retrieved and
simulated daily AOD averaged in NCP from 05 December 2015 to 04 January 2016.




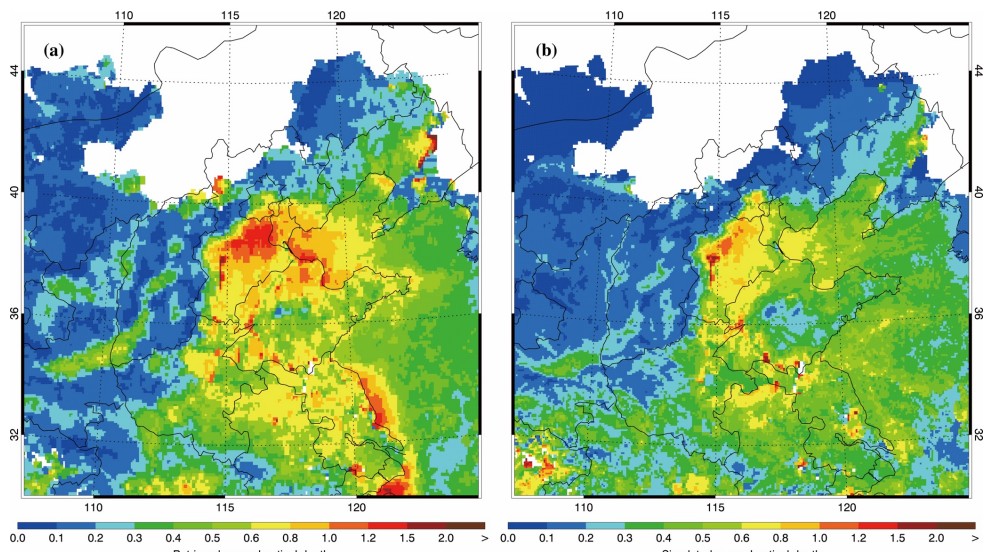

**994**
**995**
**996** Figure 6 Spatial distribution of (a) retrieved and (b) simulated AOD averaged from 05
**997** December 2015 to 04 January 2016 in NCP.
**998**
**999**
**1000**
**1001**
**1002**





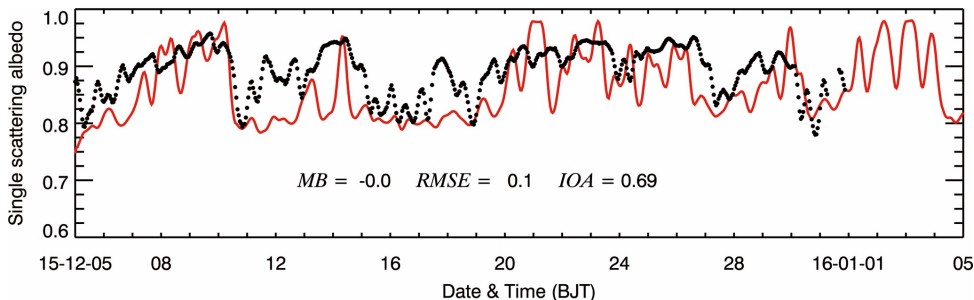

Figure 7 Comparison of measured (black dots) and predicted (red line) diurnal profiles of
SSA in Beijing from 05 December 2015 to 04 January 2016.





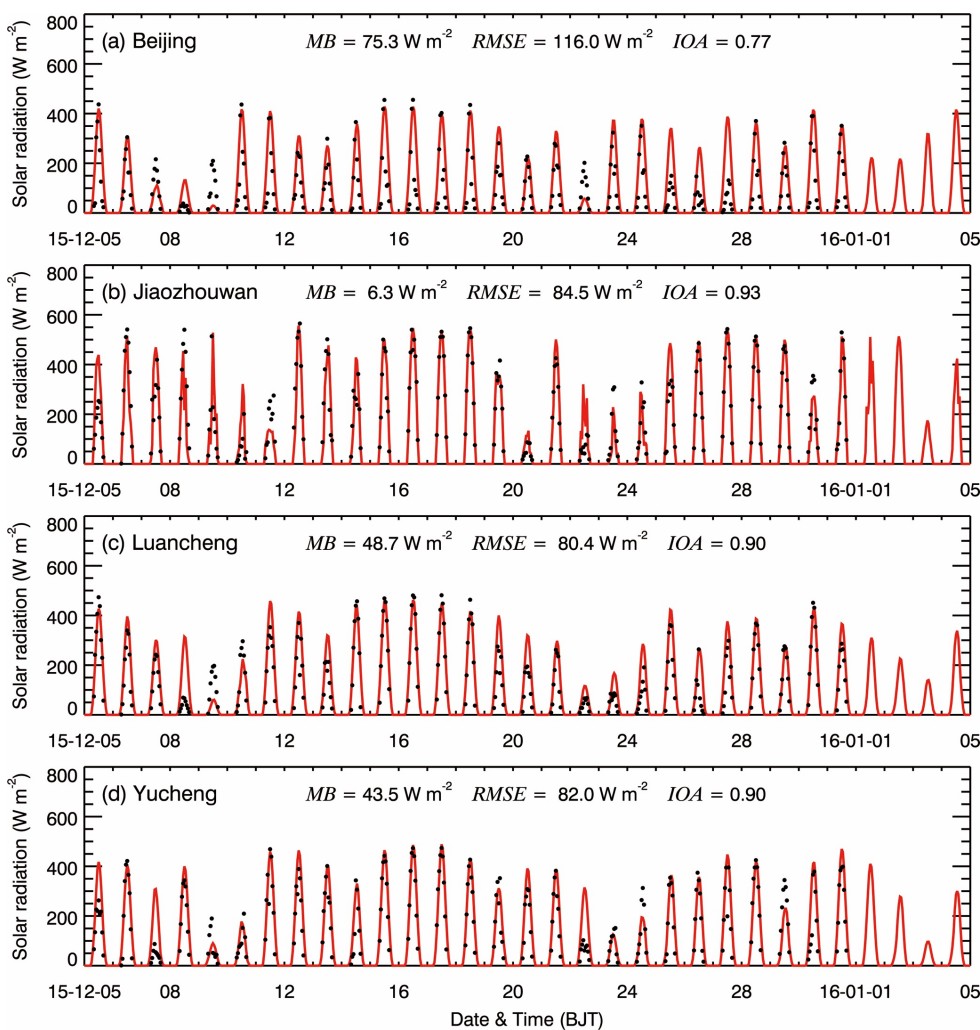

Figure 8 Comparison of measured (black dots) and predicted (red line) diurnal profiles of the
SWDOWN reaching the ground surface in (a) Beijing, (b) Jiaozhouwan, (c) Luancheng, and
(d) Yucheng from 05 December 2015 to 04 January 2016.





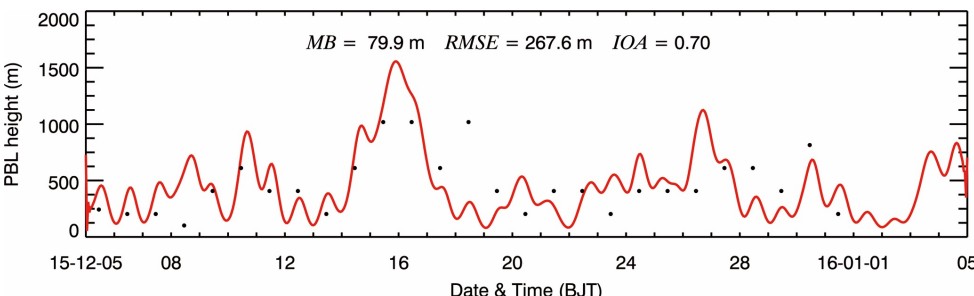

Figure 9 Comparison of predicted diurnal profile (red line) of PBLH from 05 December 2015
to 04 January 2016 with observations at 12:00 BJT in Beijing.



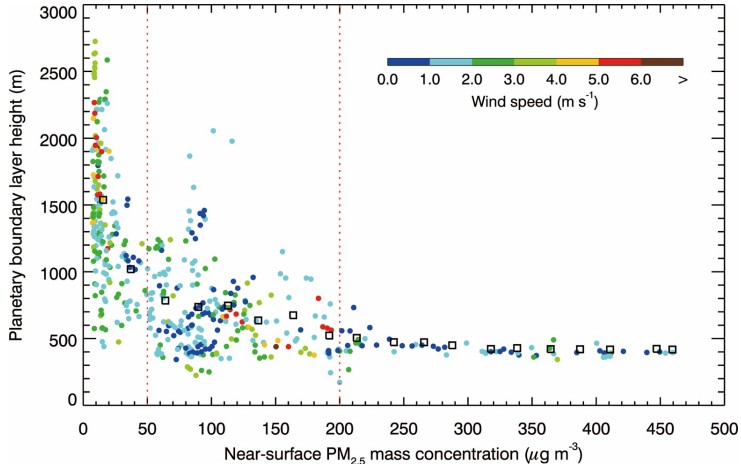

Figure 10 Scatter plot of the PBLH and near-surface [PM$_{2.5}$] at IRSDE site from 12 January
to 20 February 2014. The black rectangle shows the bin average of PBLH. The color of the
filled circles denotes the WSPD at the meteorological site close to IRSDE in Figure 1b.





Figure 11 Temporal variations of the average (a) near-surface [$PM_{2.5}$], (b) SWDOWN at the ground surface, (c) TSFC, (d) PBLH, and (e) RH in the most polluted area in NCP with [$PM_{2.5}$] of more than 150 µg m$^{-3}$ in **f$_{base}$** (red solid line) and **f$_{rad0}$** (blue solid line) from 05 December 2015 to 04 January 2016.





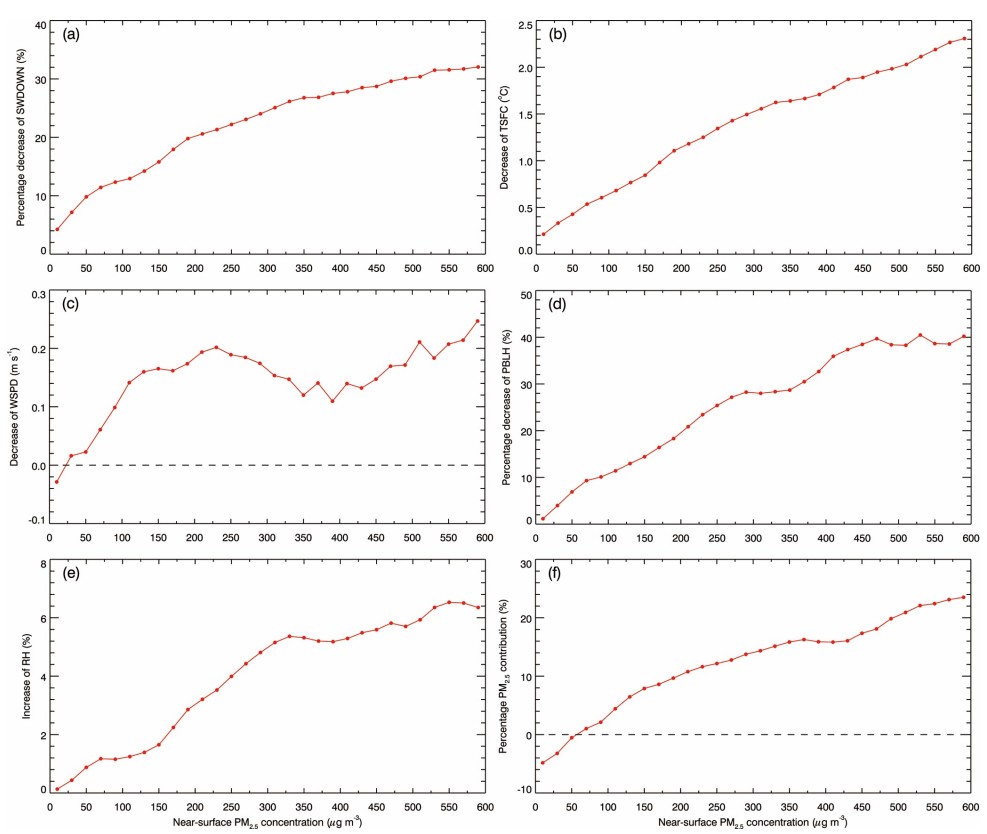



Figure 12 Average (a) percentage decrease of SWDOWN at the ground surface, (b) decrease
of TSFC, (c) decrease of WSPD, (d) percentage decrease of PBLH, (e) increase of RH, and (f)
percentage contribution of near-surface [PM$_{2.5}$] caused by the ARF, as a function of the
near-surface [PM$_{2.5}$] in NCP during daytime from 05 December 2015 to 04 January 2016.







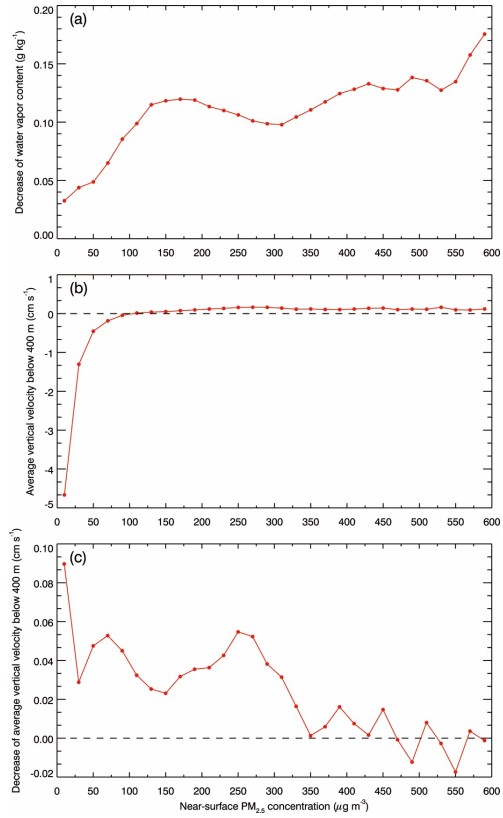

Figure 13 Average (a) decrease of water vapor content and (c) increase of average vertical
velocity below 400 m caused by the ARF, and (b) average vertical velocity below 400 m as a
function of the near-surface [PM$_{2.5}$] in NCP during daytime from 05 December 2015 to 04
January 2016.






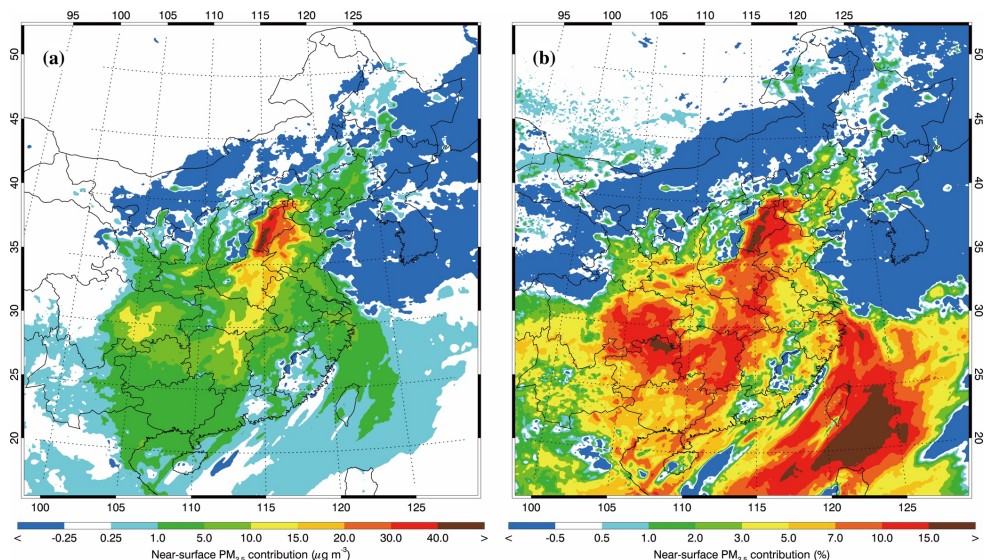

Figure 14 Near-surface [PM$_{2.5}$] contribution caused by the ARF, averaged from 05 December 2015 to 04 January 2016 in NCP.



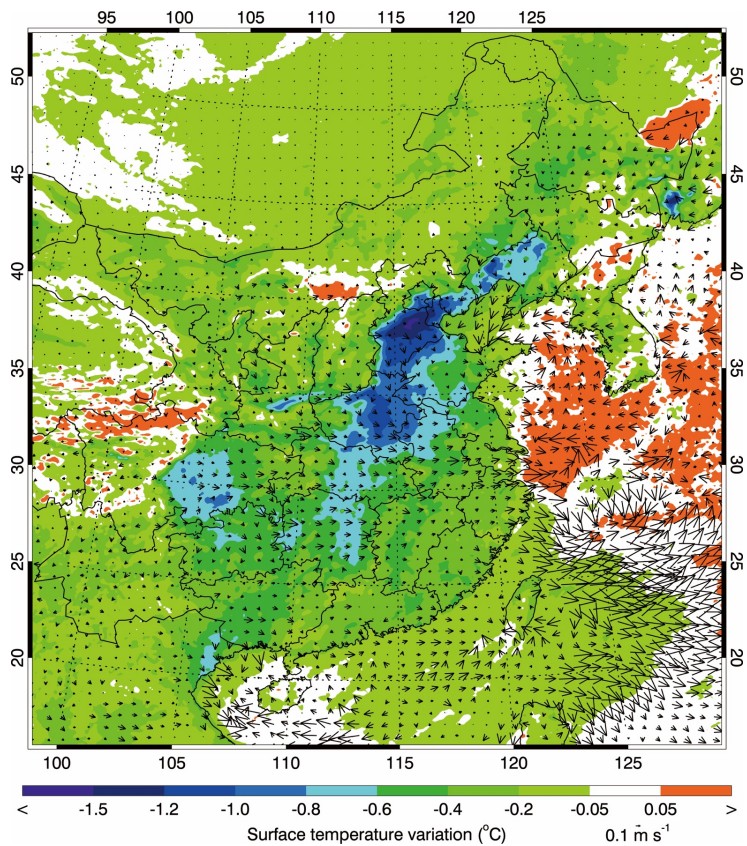

Figure 15 TSFC and wind filed variations caused by the ARF, averaged from 05 December 2015 to 04 January 2016 in NCP.