# Peer review of "Aerosol-radiation feedback deteriorates the wintertime haze in North China Plain"

_Atmospheric Chemistry and Physics, 2018_

## Referee Comment (RC1) · Anonymous Referee #1 · 20 Feb 2019

In this study, authors used WRF-Chem model to simulate a heavy haze pollution episode from 05 December 2015 to 04 January 2016 in the North China Plain (NCP) to study contributions of the aerosol shortwave radiative feedback (ARF) to near-surface PM2.5 mass concentrations. The topic is within ACP scope. Although such studies have been done for Europe (Forkel et al., 2012) or Eastern China (Zhang et al., 2015), this study focused a high PM2.5 event at the NCP. I would like to see my below comments well addressed before the official publication of the manuscript in ACP. Forkel, R., Werhahn, J., Hansen, A. B., McKeen, S., Peckham, S., Grell, G., and Suppan, P.: Effect of aerosol-radiation feedback on regional air quality–A case study with WRF/Chem, Atmospheric environment, 53, 202-211, 2012. Zhang, B., Wang, Y., and Hao, J.: Simulating aerosol–radiation–cloud feedbacks on meteorology and air quality

over eastern China under severe haze conditionsin winter, Atmospheric Chemistry and Physics, 15, 2387-2404, 2015. Comments: 1. The paper has certain unprofessional usages that hinder the reading, examples from Abstract are: a. WRF-Chem is the official name, avoid using WRF-CHEM b. Line 18: Atmospheric aerosols are different from fine particulate matters c. Line 30, there are two "during the episode" separated by a "." d. Line 34, how do you "cooling the temperature" e. Lines 32-36, this sentence has mixed verb tenses, not clear what leads to leads. f. In Line 32, "Sensitivity studies have revealed" while in Line 37 "ensemble analysis indicates" g. How "near-surface" is defined? h. I don't think "the" is needed in front of ARF, but it is needed in front of NCP i. Do not use [PM2.5] j. Line 98, correct the WRF name– the Weather Research and Forecasting (WRF) 2. What is the definition of haze pollution? I understand the authors want to say high PM2.5 concentration. But is haze pollution some well-defined concept, any criteria to that? 3. What version of WRF-Chem is used? We do appreciate the authors' efforts on improving the model, but the WRF-Chem has been developed much further since 2005 version. How did the authors incorporate the new features of the new versions? The authors also need provide reasons why the old things are used when new versions have been out for many years for the parts they modified, i.e., CMAQ aerosol module (AERO5 or AERO6?), ISORROPIA 1.7 as ISORROPIA II has been out since 2007. 4. Lines 176-182, put these equations to appendix. 5. What is the difference between summary and conclusions? Reduce the length of this section, do not repeat the main results. 6. The results are normal and well described, although the main findings (Figures 11-15) are much less compared to the model validation figures (1-10). It is suggested to consider adjusting that if ACP is sensitive to the length of articles.

---

## Referee Comment (RC2) · Anonymous Referee #2 · 8 May 2019

The authors attempt to investigate the effect of aerosol-radiation feedback (ARF) on aerosol pollution at surface by using modeling simulations. The performance of WRF-CHEM simulations were fully evaluated, and the contribution of aerosol-radiation feedback to the near-surface PM2.5 concentration was carefully quantified. However, I still have some minor issues about this work prior to its publication.

1. There are several problems about how the authors explain why ARF shows a negative effect on surface PM2.5 concentration when PM2.5 is less than 50 ug/m3. I understand that the suppressed updrafts result in less PM2.5 at surface, but I don't think it is the case that the enhanced downward motion leads to reduction in PM2.5 at surface (lines 476-578)? Also, what is the vertical velocity in Fig. 13 referring to, up-drafts, downdrafts, or the net velocity by combining updrafts and downdrafts? Is panel

(b) for the simulation of base case? The Y-axis label of (c) panel is different from the description in figure caption.

2. This work primarily quantifies to what extent the surface PM2.5 could be enhanced because of the collapse of PBL when ARF is considered. How about the impacts of ARF on AOD, which can be used to denote the column-integrated aerosol abundance? The reason why I am care about how the AOD changes under ARF effect is because the reduced incoming solar radiation might suppress the photochemical formation of PM, which could offset the effect of PBL collapse.

3. Relative to the sensitivity study section, the evaluations of model performance appear as the major portion of the body text. The authors might want to shorten the model evaluation section a little bit, so that the entire manuscript looks more balance.

---

## Author Comment (AC1) · 19 Jun 2019

**Reply to Anonymous Referee #1**

We thank the reviewer for the careful reading of the manuscript and helpful comments. We have revised the manuscript following the suggestion, as described below.

In this study, authors used WRF-Chem model to simulate a heavy haze pollution episode from 05 December 2015 to 04 January 2016 in the North China Plain (NCP) to study contributions of the aerosol shortwave radiative feedback (ARF) to near-surface $PM_{2.5}$ mass concentrations. The topic is within ACP scope. Although such studies have been done for Europe (Forkel et al., 2012) or Eastern China (Zhang et al., 2015), this study focused a high $PM_{2.5}$ event at the NCP. I would like to see my below comments well addressed before the official publication of the manuscript in ACP. Forkel, R., Werhahn, J., Hansen, A. B., McKeen, S., Peckham, S., Grell, G., and Suppan, P.: Effect of aerosol-radiation feedback on regional air quality-A case study with WRF/Chem, Atmospheric environment, 53, 202-211, 2012. Zhang, B., Wang, Y., and Hao, J.: Simulating aerosol-radiation-cloud feedbacks on meteorology and air quality over eastern China under severe haze conditions in winter, Atmospheric Chemistry and Physics, 15, 2387-2404, 2015.

**Response:** We thank the reviewer for the helpful comment and have clarified in Introduction: "*Online-coupled meteorology and chemistry models have also been used to verify the impact of ARF on the PBLH and near-surface [PM$_{2.5}$] during haze episodes in Europe, Eastern China and Northern China (Forkel et al., 2012; Z. Wang et al., 2014; Wang et al., 2015; Zhang et al., 2015; Gao et al., 2015). However, the ARF impact on near-surface [PM$_{2.5}$] varies, depending on the evaluation time and location (Table 1).*". We have also included the results of Zhang et al. (2015) into Table 1.

**1 Comment**: The paper has certain unprofessional usages that hinder the reading, examples from Abstract are:

**a.** WRF-Chem is the official name, avoid using WRF-CHEM

**b.** Line 18: Atmospheric aerosols are different from fine particulate matters

**c.** Line 30, there are two "during the episode" separated by a "."

**d**. Line 34, how do you "cooling the temperature"

**e.** Lines 32-36, this sentence has mixed verb tenses, not clear what leads to leads.

**f.** In Line 32, "Sensitivity studies have revealed" while in Line 37 "ensemble analysis

indicates"

**g.** How "near-surface" is defined?

**h.** I don't think "the" is needed in front of ARF, but it is needed in front of NCP

**i.** Do not use [PM$_{2.5}$]

**j.** Line 98, correct the WRF name-the Weather Research and Forecasting (WRF)

**Response:**

**a.** We have revised the "WRF-CHEM" as "WRF-Chem" in the manuscript.

**b.** We have removed "*fine particulate matters (PM$_{2.5}$)*" in Abstract.

**c.** We have removed the first "during the episode".

**d.** We have changed "*cooling the temperature of the low-level atmosphere*" as " to *cool the low-level atmosphere*".

**e.** We have rephrased the sentence as "*Sensitivity studies have revealed that high loadings of PM$_{2.5}$ attenuate the incoming solar radiation reaching the surface to cool the low-level atmosphere, suppressing development of PBL, decreasing the surface wind speed, further hindering the PM$_{2.5}$ dispersion and consequently exacerbating the haze pollution in the NCP.*"

**f.** We have changed "*The ensemble analysis indicates*" to "*Furthermore*".

**g.** We have clarified in Abstract: "*near-surface (around 15 m above the ground surface)*".

**h.** We have revised "*the ARF*" as "*ARF*" and "*NCP*" as "*the NCP*" in the manuscript.

**i.** We have defined "PM$_{2.5}$ concentrations" as "[PM$_{2.5}$]" for convenience in Section 1.

**j.** We have corrected the WRF name as "*the Weather Research and Forecasting (WRF)*" in Section 1.

**2 Comment:** What is the definition of haze pollution? I understand the authors want to say high PM$_{2.5}$ concentration. But is haze pollution some well-defined concept, any criteria to that?

**Response**: We have clarified in Section 2.1: "*During the study episode, the average hourly [PM$_{2.5}$] in the NCP are approximately 127.9 µg m$^{-3}$, within the fourth grade of National Ambient Air Quality Standards with [PM$_{2.5}$] between 115 and 150 µg m$^{-3}$ (moderately polluted, Feng et al., 2016). The persistent and widespread haze pollution episode with high [PM$_{2.5}$] in the NCP provides a suitable case for observation analyses and model simulations to investigate ARF effect on haze pollution.*".

**3 Comment:** What version of WRF-Chem is used? We do appreciate the authors' efforts on improving the model, but the WRF-Chem has been developed much further since 2005 version. How did the authors incorporate the new features of the new versions? The authors also need provide reasons why the old things are used when new versions have been out for many years for the parts they modified, i.e., CMAQ aerosol module (AERO5 or AERO6?), ISORROPIA 1.7 as ISORROPIA II has been out since 2007.

**Response:** We have clarified in Section 3.3:

"*The WRF-Chem model (Grell et al., 2005) with modifications by Li et al. (2010, 2011a, b, 2012) is applied to evaluate effects of ARF on the wintertime haze formation in the NCP. The model includes a new flexible gas phase chemical module, which can be used with different chemical mechanisms, such as CBIV, RADM2, and SAPRC. In the study, the SAPRC99 chemical mechanism is used based on the available emission inventory. For the aerosol simulations, the CMAQ/models3 aerosol module (AERO5) developed by US EPA has been incorporated into the model (Binkowski and Roselle, 2003). The wet deposition is based on the method in the CMAQ module and the dry deposition of chemical species follows Wesely (1989). The photolysis rates are calculated using the FTUV (fast radiation transfer model) with the aerosol and cloud effects on photolysis (Li et al., 2005, 2011a).*

*It is worth noting that the most recent extension of ISORROPIA, known as ISORROPIA II, has incorporated a larger number aerosol species (Ca, Mn, K salts) and is designed to be a superset of ISORROPIA (Fountoukis et al., 2009). However, the ISORROPIA Version II uses the exact same routines as ISORROPIA to compute the equilibrium composition, which produces identical results as ISORROPIA when crustal species are not considered. Therefore, the inorganic aerosols in this study are predicted using ISORROPIA Version 1.7, calculating the composition and phase state of an ammonium-sulfate-nitrate-water inorganic aerosol in thermodynamic equilibrium with gas phase precursors (Nenes, 1998). In addition, a parameterization of sulfate heterogeneous formation involving aerosol liquid water (ALW) has been developed and implemented into the model, which has successfully reproduced the observed rapid sulfate formation during haze days (Li et al., 2017a). The sulfate heterogeneous formation from $SO_2$ is parameterized as a first order irreversible uptake by ALW surfaces, with a reactive uptake coefficient of $0.5 \times 10^{-4}$ assuming that there is enough alkalinity to maintain the high iron-catalyzed reaction rate.*

*The OA module is based on the VBS approach with aging and detailed information can be found in Li et al. (2011b). The POA components from traffic-related combustion and*

*biomass burning are represented by nine surrogate species with saturation concentrations (C\*) ranging from $10^{-2}$ to $10^6$ μg m$^{-3}$ at room temperature (Shrivastava et al., 2008), and assumed to be semi-volatile and photochemically reactive (Robinson et al., 2007). The SOA formation from each anthropogenic or biogenic precursor is calculated using four semi-volatile VOCs with effective saturation concentrations of 1, 10, 100, and 1000 μg m$^{-3}$ at 298 K. The SOA formation via the heterogeneous reaction of glyoxal and methylglyoxal is parameterized as a first-order irreversible uptake by aerosol particles with an uptake coefficient of $3.7\times10^{-3}$ (Liggio et al., 2005; Zhao et al., 2006; Volkamer et al., 2007).".*

**4 Comment:** Lines 176-182, put these equations to appendix.

**Response:** We have moved these equations to the Supplement Section S1.

**5 Comment:** What is the difference between summary and conclusions? Reduce the length of this section, do not repeat the main results.

**Response:** We have revised "*Summary and conclusions*" as "*Conclusions*" in Section 4, and also reduced the length of the section.

**6 Comment:** The results are normal and well described, although the main findings (Figures 11-15) are much less compared to the model validation figures (1-10). It is suggested to consider adjusting that if ACP is sensitive to the length of articles.

**Response:** We have moved the model validation of air pollutants and aerosol species to the supplement to shorten the part of model evaluations, and clarified in Section 3.1: "*Generally, the model simulates well the horizontal distributions and temporal variations of PM$_{2.5}$, O$_3$, NO$_2$, and SO$_2$ mass concentrations against measurements in the NCP. Additionally, the model also reasonably well reproduces the temporal profiles of the aerosol species compared to observations in Beijing. The detailed model validation of air pollutants in the NCP and the aerosol species in Beijing can be found in SI.*"

**Reference**

Feng, T., Bei, N., Huang, R.-J., Cao, J., Zhang, Q., Zhou, W., Tie, X., Liu, S., Zhang, T., Su, X., Lei, W., Molina, L. T., and Li, G.: Summertime ozone formation in Xi'an and surrounding areas, China, Atmos. Chem. Phys., 16, 4323-4342, https://doi.org/10.5194/acp-16-4323-2016, 2016.

Forkel, R., Werhahn, J., Hansen, A.B., Mckeen, S., Peckham, S., Grell, G., Suppan, P.: Effect of aerosol-radiation feedback on regional air quality - A case study with WRF/Chem, Atmos. Environ., 53, 202-211, 2012.

Fountoukis, C., Nenes, A., Sullivan, A., Weber, R., VanReken, T., Fischer, M., Matias, E., Moya, M. Farmer, D., and Cohen, R.: Thermodynamic characterization of Mexico City Aerosol during MILAGRO 2006, Atmos. Chem. Phys., 9, 2141–2156, 2009.

Li, G., Bei, N., Cao, J., Huang, R., Wu, J., Feng, T., Wang, Y., Liu, S., Zhang, Q., Tie, X., and Molina, L. T.: A possible pathway for rapid growth of sulfate during haze days in China, Atmos. Chem. Phys., 17, 3301–3316, https://doi.org/10.5194/acp-17-3301-2017, 2017.

Li, G., Bei, N., Tie, X., and Molina, L. T.: Aerosol effects on the photochemistry in Mexico City during MCMA-2006/MILAGRO campaign, Atmos. Chem. and Phys., 11, 5169–5182, https://doi.org/10.5194/acp-11-5169-2011, 2011a.

Li, G., Lei, W., Bei, N., and Molina, L. T.: Contribution of garbage burning to chloride and $PM_{2.5}$ in Mexico City, Atmos. Chem. and Phys. 12, 8751–8761, https://doi.org/10.5194/acp-12-8751-2012, 2012.

Li, G., Lei, W., Zavala, M., Volkamer, R., Dusanter, S., Stevens, P., and Molina, L. T.: Impacts of HONO sources on the photochemistry in Mexico City during the MCMA-2006/MILAGO Campaign, Atmos. Chem. and Phys., 10, 6551–6567, https://doi.org/10.5194/acp-10-6551-2010, 2010.

Li, G., Zavala, M., Lei, W., Tsimpidi, A. P., Karydis, V. A., Pandis, S. N., Canagaratna, M. R., and Molina, L. T.: Simulations of organic aerosol concentrations in Mexico City using the WRF-Chem model during the MCMA-2006/MILAGRO campaign, Atmos. Chem. and Phys., 11, 3789–3809, https://doi.org/10.5194/acp-11-3789-2011, 2011b.

Li, G., Zhang, R., Fan, J., and Tie, X.: Impacts of black carbon aerosol on photolysis and ozone, J. Geophys. Res.-Atmos., 110, D23206, https://doi.org/10.1029/2005JD005898, 2005.

Liggio, J., Li, S. M., and McLaren, R.: Reactive uptake of glyoxal by particulate matter, J. Geophys. Res.-Atmos., 110, doi: 10.1029/2004jd005113, 2005.

Robinson, A. L., Donahue, N. M., Shrivastava, M. K.,Weitkamp, E. A., Sage, A. M., Grieshop, A. P., Lane, T. E., Pandis, S. N., and Pierce, J. R.: Rethinking organic aerosols: semivolatile emissions and photochemical aging, Science, 315, 1259–1262, 2007.

Shrivastava, M. K., Lane, T. E., Donahue, N. M., Pandis, S. N., and Robinson, A. L.: Effects of gas particle partitioning and aging of primary emissions on urban and regional

organic aerosol concentrations, J. Geophys. Res.-Atmos., 113, doi: 10.1029/2007jd009735, 2008.

Volkamer, R., Martini, F. S., Molina, L. T., Salcedo, D., Jimenez, J. L., and Molina, M. J.: A missing sink for gas-phase glyoxal in Mexico City: Formation of secondary organic aerosol, Geophys. Res. Lett., 34, doi: 10.1029/2007gl030752, 2007.

Zhang, B., Wang, Y., and Hao, J.: Simulating aerosol-radiation-cloud feedbacks on meteorology and air quality over eastern China under severe haze conditions in winter, Atmos. Chem. Phys., 15, 2387-2404, 10.5194/acp-15-2387-2015, 2015.

Zhao, J., Levitt, N. P., Zhang, R., and Chen, J.: Heterogeneous reactions of methylglyoxal in acidic media: Implications for secondary organic aerosol formation, Environ. Sci. Technol., 40, 7682–7687, doi: 10.1021/es060610k, 2006.

---

## Author Comment (AC2) · 19 Jun 2019

**Reply to Anonymous Referee #2**

We thank the reviewer for the careful reading of the manuscript and helpful comments. We have revised the manuscript following the suggestion, as described below.

The authors attempt to investigate the effect of aerosol-radiation feedback (ARF) on aerosol pollution at surface by using modeling simulations. The performance of WRF-CHEM simulations was fully evaluated, and the contribution of aerosol-radiation feedback to the near-surface $PM_{2.5}$ concentration was carefully quantified. However, I still have some minor issues about this work prior to its publication.

**1 Comment**: There are several problems about how the authors explain why ARF shows a negative effect on surface $PM_{2.5}$ concentration when $PM_{2.5}$ is less than 50 ug/m$^3$. I understand that the suppressed updrafts result in less $PM_{2.5}$ at surface, but I don't think it is the case that the enhanced downward motion leads to reduction in $PM_{2.5}$ at surface (lines 476-578)? Also, what is the vertical velocity in Fig. 13 referring to, updrafts, downdrafts, or the net velocity by combining updrafts and downdrafts? Is panel (b) for the simulation of base case? The Y-axis label of (c) panel is different from the description in figure caption.

**Response:** We have clarified in Section 3.3: "*Figure 11b presents the average vertical velocity (the net velocity by combining updrafts and downdrafts) below about 400 m in $f_{rad0}$ as a function of near-surface [PM$_{2.5}$]. Apparently, when ARF is not considered, the area with near-surface [PM$_{2.5}$] less than 100 μg m$^{-3}$ is generally controlled by downward airflow, and vice versa for the area with near-surface [PM$_{2.5}$] more than 100 μg m$^{-3}$. The ARF induced cooling generally cause a downward motion in the PBL (Figure 11c), which suppresses the upward motion in the area with near-surface [PM$_{2.5}$] more than 100 μg m$^{-3}$ to enhance near-surface [PM$_{2.5}$], but accelerates the downward motion in the area with near-surface [PM$_{2.5}$] less than 100 μg m$^{-3}$ to strengthen the divergence intensity, further decreasing near-surface [PM$_{2.5}$].*". We have revised the figure caption "*Figure 13 Average (a) decrease of water vapor content and (c) increase of average vertical velocity below 400 m caused by ARF, and (b) average vertical velocity below 400 m as a function of the near-surface [PM$_{2.5}$] in NCP during daytime from 05 December 2015 to 04 January 2016.*" as "*Figure 11 Average decrease of (a) near-surface water vapor content and (c) vertical velocity below 400 m caused by ARF, and (b) average vertical velocity below 400 m in $f_{rad0}$ as a function of the*

*near-surface [PM₂.₅] in the NCP during daytime from 05 December 2015 to 04 January 2016.*"

**2 Comment:** This work primarily quantifies to what extent the surface PM₂.₅ could be enhanced because of the collapse of PBL when ARF is considered. How about the impacts of ARF on AOD, which can be used to denote the column-integrated aerosol abundance? The reason why I care about how the AOD changes under ARF effect is because the reduced incoming solar radiation might suppress the photochemical formation of PM, which could offset the effect of PBL collapse.

**Response:** We have clarified in Section 3.3 "*Figure 9 presents the temporal variation of AOD at 550nm averaged in the selected area during the episode in $f_{base}$ and $f_{rad0}$ to evaluate the impact of ARF on AOD. Apparently, except from 8 to 11 December, the ARF contribution to AOD is generally marginal, indicating that ARF does not play an important role in the column-integrated aerosol abundance. Additionally, the considerable AOD enhancement from 8 to 11 December is more likely caused by the substantial increase in RH due to ARF, which facilitates aerosol hygroscopic growth to augment particle size and further increases AOD. It is worth noting that the extinction of haze aerosols in the PBL also decreases the photolysis to suppress the photochemistry, further hindering the secondary aerosol formation to offset effects of ARF on near-surface [PM₂.₅].*"

We have also clarified in Section 4: "*It is worth noting that modification of photolysis by aerosol scattering or absorbing solar radiation ultimately alters the atmospheric oxidizing capacity to influence the secondary aerosol formation, which potentially offsets ARF effect on the haze pollution. Hence, further studies need to be performed to evaluate the effect of aerosol photolysis interaction on the haze pollution.*".

**3 Comment:** Relative to the sensitivity study section, the evaluations of model performance appear as the major portion of the body text. The authors might want to shorten the model evaluation section a little bit, so that the entire manuscript looks more balance.

**Response:** We have moved the model validation of air pollutants and aerosol species to the supplement to shorten the part of model evaluations, and clarified in Section 3.1: "*Generally, the model simulates well the horizontal distributions and temporal variations of PM₂.₅, O₃, NO₂, and SO₂ mass concentrations against measurements in the NCP. Additionally, the*

*model also reasonably well reproduces the temporal profiles of the aerosol species compared to observations in Beijing. The detailed model validation of air pollutants in the NCP and the aerosol species in Beijing can be found in SI.''*